# Object-Centric World Models from Few-Shot Annotations for Sample-Efficient Reinforcement Learning

**Weipu Zhang**[◇†⋆⋆]   **Adam Jelley**[†]   **Trevor McInroe**[†]   **Amos Storkey**[†]   **Gang Wang**[◇‡]

[◇]National Key Lab of Autonomous Intelligent Unmanned Systems, Beijing Institute of Technology
[†]University of Edinburgh   [⋆]Jiangxing Intelligence Inc.
`weipuzhang.academic@gmail.com`
`{adam.jelley,t.mcinroe,a.storkey}@ed.ac.uk`
`gangwang@bit.edu.cn`

## Abstract

While deep reinforcement learning (RL) from pixels has achieved remarkable success, its sample inefficiency remains a critical limitation for real-world applications. Model-based RL (MBRL) addresses this by learning a world model to generate simulated experience, but standard approaches that rely on pixel-level reconstruction losses often fail to capture small, task-critical objects in complex, dynamic scenes. We posit that an object-centric (OC) representation can direct model capacity toward semantically meaningful entities, improving dynamics prediction and sample efficiency. In this work, we introduce **OC-STORM**, an object-centric MBRL framework that enhances a learned world model with object representations extracted by a pretrained segmentation network. By conditioning on a minimal number of annotated frames, OC-STORM learns to track decision-relevant object dynamics and inter-object interactions without extensive labeling or access to privileged information. Empirical results demonstrate that OC-STORM significantly outperforms the STORM baseline on the Atari 100k benchmark and achieves state-of-the-art sample efficiency on challenging boss fights in the visually complex game **Hollow Knight**. Our findings underscore the potential of integrating OC priors into MBRL for complex visual domains.
Project page: **https://oc-storm.weipuzhang.com**

## 1 Introduction

Deep reinforcement learning (RL) has achieved landmark successes in domains ranging from board games to robotic control (Silver et al., 2016; Mnih et al., 2015; Hafner et al., 2023; Feng et al., 2026). However, a fundamental limitation persists: Deep RL agents are notoriously sample-inefficient, often requiring orders of magnitude more experience than humans to master a task. This inefficiency is particularly pronounced in pixel-based environments, where the agent must learn both visual representations and control policies from high-dimensional observations.

Model-based RL (MBRL) offers a promising path toward greater sample efficiency by learning a predictive model of the environment dynamics (Sutton & Barto, 2018; Ha & Schmidhuber, 2018). Contemporary MBRL methods typically learn this world model in a self-supervised manner, using autoregressive prediction trained with pixel-wise reconstruction losses (e.g., the $\ell_2$-loss) (Hafner et al., 2023; Zhang et al., 2023; Micheli et al., 2023). While effective in many cases, this approach has a critical weakness: the reconstruction objective is dominated by large, static background elements, often at the expense of neglecting small, sparse, yet decision-critical objects. As illustrated in Figure 1, standard world models like STORM can accurately reconstruct the background of a visually complex game like Hollow Knight but fail to capture essential elements such as the boss character, which is key for successful policy learning, leading to poor performance.

---

[⋆]Work was initiated at the University of Edinburgh and completed at the Beijing Institute of Technology.
[‡]Corresponding author.

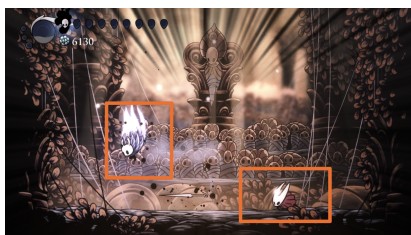

(a) A high-resolution sample observation from Hollow Knight.

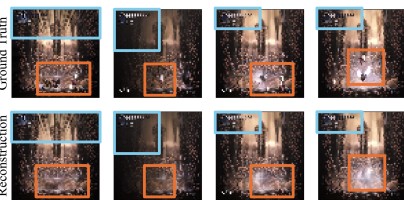

(b) Reconstruction generated by a trained STORM agent.

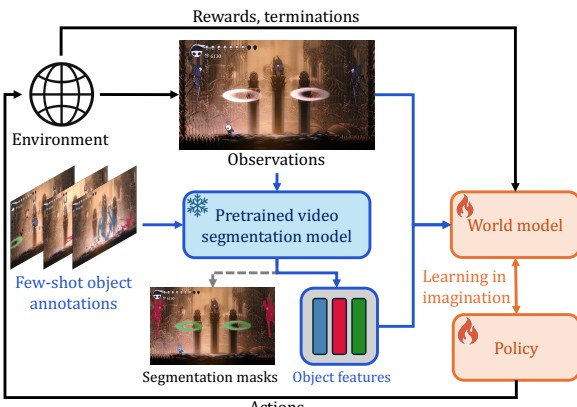

(c) The proposed OC-STORM framework. A frozen, pretrained segmentation model extracts object feature vectors from a few annotated frames. These features are combined with downsampled pixels to train an OC world model, which is then used for policy learning via imagined trajectories.

Figure 1: Left: STORM (Zhang et al., 2023) accurately reconstructs large background areas (blue) but overlooks the small, critical player and boss characters (orange), hindering policy learning. Right: Overview of the proposed OC-STORM framework. See Appendix A for network details.

A natural solution is to incorporate object-level inductive biases, guiding the world model to represent the environment in terms of discrete, interacting entities. Historically, this required extensive task-specific labeling, limiting its practicality. Recent advances in open-set segmentation and tracking models, including SAM (Kirillov et al., 2023; Ravi et al., 2024), Cutie (Cheng et al., 2023), and GroundingDINO (Liu et al., 2023), have changed this landscape. These foundation models can now generate high-quality object segmentations for novel domains with only a handful of annotations, making OCRL a viable and powerful approach.

In this work, we introduce **OC-STORM**, an object-centric MBRL framework that enhances world models with object representations extracted by a pretrained segmentation network (Figure 1c). Our approach begins by annotating key objects in a handful of (e.g., $6-12$) frames. A frozen, pretrained video segmentation network (e.g., Cutie (Cheng et al., 2023)) then extracts compact feature vectors for these objects, which are fused with raw pixel observations to train a world model that explicitly reasons about object dynamics and inter-object interactions. The policy is subsequently trained on imagined trajectories from the world model. By focusing model capacity on task-relevant entities, OC-STORM improves sample efficiency without requiring extensive labeling or access to internal environment states.

Our main contributions are summarized as follows.

c1) We propose OC-STORM, a novel MBRL framework which, to our knowledge, is **the first to successfully integrate few-shot, pretrained object segmentation models into world models for both the Atari** 100**k benchmark and the visually complex game Hollow Knight**. This general applicability is achieved without extensive labeling or access to internal state information.

c2) We provide a comprehensive empirical evaluation across diverse domains (Atari, Hollow Knight), model backbones (STORM, DreamerV3), and segmentation methods (Cutie, SAM2). Our results show that OC-STORM achieves state-of-the-art sample efficiency, particularly in environments where key information is localized in objects.

c3) We conduct an extensive ablation study that yields practical insights for future OCRL methods, including comparisons between vector-based and mask-based object representations and an analysis of robustness to segmentation errors.

## 2 PRELIMINARIES

We formulate the game control problem as a finite-horizon Markov decision process (MDP) $\mathcal{M} = \langle \mathcal{S}, \mathcal{A}, p, r, \gamma \rangle$, with latent state space $\mathcal{S}$, action space $\mathcal{A}$, transition kernel $p(s_{t+1}|s_t, a_t)$, reward function $r : \mathcal{S} \times \mathcal{A} \rightarrow \mathbb{R}$, and discount factor $\gamma \in [0, 1)$. Our objective is to learn a policy $\pi_\theta(a_t|s_t)$ parameterized by $\theta$ that maximizes the expected return $\mathbb{E}_{\pi_\theta, p}\left[\sum_{t=0}^{T-1} \gamma^t r_t\right]$. We adopt a standard MBRL paradigm: first, learn a world model $p_\phi$ via self-supervised learning on environment interactions $\{o_t, a_t, r_t\}$; then, train the policy $\pi_\theta$ using trajectories generated by the model.

To extract object information from out-of-domain environments (e.g., Atari) with minimal human effort, we evaluated several state-of-the-art object segmentation, detection, and tracking methods (Cheng & Schwing, 2022; Cheng et al., 2023; Kirillov et al., 2023; Zhang et al., 2024; Ravi et al., 2024; Redmon et al., 2016; Jocher et al., 2023; Liu et al., 2023; Locatello et al., 2020; Kipf et al., 2022; Elsayed et al., 2022; Wang et al., 2023; Xu et al., 2023); see Appendix D for a detailed review. Among these, we selected Cutie (Cheng et al., 2023) and SAM2 (Ravi et al., 2024) as our object feature extractors due to their suitability for our setting. These methods exhibit four key properties that are critical for our task:

- **Temporal consistency and efficiency.** As video-based segmentation methods, they produce consistent object representations across frames and support real-time processing of high-resolution inputs on a single consumer-grade GPU.
- **Retrieval-based flexibility.** They effectively utilize few-shot annotations through retrieval from a memory bank, which is crucial in complex environments where a single frame or natural language description may not capture all object states.
- **Compact object representations.** They provide compact vector embeddings that preserve sufficient semantic and visual information necessary for decision-making.
- **Out-of-domain robustness.** Despite being trained on natural images without exposure to Hollow Knight or Atari frames, they generalize effectively to Atari and Hollow Knight frames, as demonstrated in our experiments.

Both SAM2 and Cutie employ an **object-attention** mechanism that facilitates object-level and visual-level information interaction. This design leverages high-level semantic information to improve segmentation performance. We harness this mechanism to extract compact object-level feature representations for our world model.

## 3 METHOD

Our method consists of two stages: (1) self-supervised learning of an OC world model that captures environment dynamics, and (2) training an actor-critic policy using trajectories imagined by the model. We denote the world model parameters as $\phi$, the critic network parameters as $\psi$, and the actor network parameters as $\theta$. We use $L$ to represent the batch length of trajectory segments and $T$ for the episode length. The model architecture is detailed in Appendix A.

### 3.1 OBJECT FEATURE AND VISUAL INPUT

We extract object feature representations using the object-attention outputs from SAM2 or Cutie. The dimension of these object features is denoted as obj_dim ($256$ for SAM2 and $2,048$ for Cutie). For visual input, we resize the original observation to $64 \times 64$ resolution, consistent with prior work (Hafner et al., 2023; Zhang et al., 2023). Formally, at each timestep $t$, we define:

$$
\begin{aligned}
\text{Observation:} \quad & o_t \in \mathbb{R}^{3 \times H \times W}, \\
\text{Object features:} \quad & s_t^{\text{obj}} = \text{SegModel}(o_t) \in \mathbb{R}^{K \times \text{obj\_dim}}, \\
\text{Visual input:} \quad & s_t^{\text{vis}} = \text{Resize}(o_t) \in \mathbb{R}^{3 \times 64 \times 64},
\end{aligned} \tag{1}
$$

where $K$ is the number of objects, which is environment-specific and set by the user (e.g., $K = 3$ for Pong: two paddles and one ball). The segmentation model (SAM2 or Cutie) maintains internal states across frames to ensure tracking consistency, which we reset at the start of each episode.

## 3.2 Latent State Discretization via Categorical VAE

Autoregressive sequence models operating directly on high-dimensional inputs are susceptible to compounding prediction errors (Hafner et al., 2023; Zhang et al., 2023). To mitigate this, we employ a categorical VAE (Kingma & Welling, 2014; Hafner et al., 2023) that encodes inputs into discrete latent representations. The encoder $q_\phi$ maps input states $s_t$ to discrete latent variables $z_t$, while the decoder $p_\phi$ reconstructs the inputs from these latents:

$$\begin{aligned} \text{Encoder:} \quad & z_t \sim q_\phi(z_t|s_t), \\ \text{Decoder:} \quad & \hat{s}_t = p_\phi(z_t). \end{aligned} \tag{2}$$

We use distinct architectures for different input modalities: multi-layer perceptron (MLP) encoders/decoders for object feature vectors ($\mathbb{R}^{K \times d_{\text{obj}}} \leftrightarrow \mathbb{R}^{K \times 16 \times 16}$) and convolutional neural network (CNN)-based ones for visual observations ($\mathbb{R}^{3 \times 64 \times 64} \leftrightarrow \mathbb{R}^{32 \times 32}$). The object latents use 16 categorical distributions with 16 classes each, while visual latents use 32 distributions with 32 classes, following established configurations (Hafner et al., 2023; Zhang et al., 2023). The decoder architectures are symmetric to the encoders. The reduced dimensionality for objects reflects their lower information content compared to full scenes. We use the straight-through gradient estimator (Bengio et al., 2013) to enable differentiable sampling from the categorical distributions.

## 3.3 Spatial-Temporal Object-Centric Dynamics

Our framework supports both transformer-based (STORM) and recurrent neural network (RNN)-based (DreamerV3) world model backbones. The core innovation is a spatial-temporal architecture that separately models object and visual dynamics while enabling interaction between them.

For the **STORM** backbone, we employ a transformer architecture with alternating spatial and temporal attention blocks. Spatial attention operates across the $K$ object tokens and one visual token at each timestep $t$: $(z_t^1, z_t^2, \ldots, z_t^K, z_t^{\text{vis}})$, capturing inter-object relationships and object-scene interactions. Temporal attention then processes each token type independently across the sequence $(z_1^i, z_2^i, \ldots, z_L^i)$ to model dynamics. Actions are incorporated via concatenation with the latent representations. For **DreamerV3**, we augment the RNN with spatial attention mechanisms at each timestep to achieve similar object-scene interaction.

The sequence model $f_\phi$ takes as input the sequence of object latents $z_{1:L}^{\text{obj}}$, visual latents $z_{1:L}^{\text{vis}}$, and actions $a_{1:L}$, and outputs hidden states $h_{1:L} \in \mathbb{R}^{(K+1) \times L \times 256}$, as

$$h_{1:L} = f_\phi(z_{1:L}^{\text{obj}}, z_{1:L}^{\text{vis}}, a_{1:L}) \in \mathbb{R}^{(K+1) \times L \times d_h} \tag{3}$$

where $d_h = 256$ denotes the hidden dimension, and $K + 1$ accounts for the $K$ object tokens plus one visual token.

## 3.4 Prediction Heads

We use the hidden states $h_t$ of the transformer to predict the next latent state, reward, and termination signal. The dynamics predictor $g_\phi^{\text{Dyn}}$ is an MLP that predicts the distribution of the next latent state $\hat{z}_{t+1}$. For reward and termination prediction, we use a self-attention mechanism that aggregates information from all objects and visual tokens, with structures detailed in Appendix A. Specifically, we introduce a special query token (similar to the [CLS] token in BERT (Devlin et al., 2019)) that attends to all tokens in $h_t$ and then passes the resulting representation through an MLP to predict the reward $\hat{r}_t$ and termination probability $\hat{\tau}_t$. These predictors are given as follows:

$$\begin{aligned} \text{Dynamics predictor:} \quad & \hat{z}_{t+1} \sim g_\phi^{\text{dyn}}(\hat{z}_{t+1}|h_t), \\ \text{Reward predictor:} \quad & \hat{r}_t = g_\phi^{\text{rew}}(h_t), \\ \text{Termination predictor:} \quad & \hat{\tau}_t = g_\phi^{\text{term}}(h_t). \end{aligned} \tag{4}$$

## 3.5 Training the World Model and Policy

We train the world model end-to-end using a combination of reconstruction and prediction losses. The world model is trained to maximize the likelihood of the observed data, including the reconstructed inputs, rewards, and terminations. The policy $\pi_\theta$ and value function $V_\psi$ are trained using

imagined trajectories generated by the world model. We use the actor-critic algorithm from DreamerV3 (Hafner et al., 2023; Zhang et al., 2023; Micheli et al., 2023; Robine et al., 2023), which involves policy gradient updates with value function baselines. Detailed loss functions and training procedures are provided in Appendix B.

## 4 EXPERIMENTS

We evaluate OC-STORM on two challenging benchmarks: the standard Atari 100k sample-efficiency benchmark (Bellemare et al., 2013) and complex boss fights from Hollow Knight (Team-Cherry, 2017), a highly acclaimed game released in 2017. This dual evaluation tests our method across diverse visual complexities—from the relatively simple but well-established Atari environments to the rich, dynamic visuals of modern games where OC reasoning provides maximum benefit.

Table 1: Game scores and overall human-normalized scores on the Atari 100k benchmark. We compare OC approaches (vector-based OC and mask-based FOCUS (Ferraro et al., 2023)) against baseline world models. Bold indicates scores within 5% of best. The highlighting is computed separately with respect to the Dreamer and STORM baselines, shown in **blue** and **red**, respectively. The last column '#obj' denotes manually annotated objects per game. All experiments use identical lightweight configurations suitable for high-resolution environments. STORM serves as the primary baseline due to its computational efficiency.

| Game | Random | Human | DreamerV3 | Vector Cutie-OC DreamerV3 | STORM | Mask FOCUS STORM | Vector SAM2-OC STORM | Vector Cutie-OC STORM | #obj |
|---|---|---|---|---|---|---|---|---|---|
| Alien | 228 | 7128 | 789 | **1234** | 748 | 1039 | 591 | **1101** | 4 |
| Amidar | 6 | 1720 | 107 | **123** | 144 | 110 | 150 | **163** | 2 |
| Assault | 222 | 742 | 928 | **1201** | **1377** | 986 | 1086 | 1270 | 4 |
| Asterix | 210 | 8503 | 916 | **1293** | 1318 | 1552 | 1408 | **1754** | 3 |
| BankHeist | 14 | 753 | **812** | **849** | 990 | 702 | 840 | **1075** | 3 |
| BattleZone | 2360 | 37188 | 7500 | **9550** | 5830 | **12870** | **12950** | 4590 | 3 |
| Boxing | 0 | 12 | 78 | **88** | 81 | 78 | 81 | **92** | 2 |
| Breakout | 2 | 30 | 39 | **44** | 41 | 57 | **61** | 52 | 3 |
| ChopperCommand | 811 | 7388 | 1926 | **2130** | 1644 | 1751 | **2580** | 2090 | 4 |
| CrazyClimber | 10780 | 35829 | 83635 | **89671** | 79196 | 65753 | 76039 | **84111** | 2 |
| DemonAttack | 152 | 1971 | 379 | **540** | 325 | 360 | **491** | 411 | 4 |
| Freeway | 0 | 30 | 0 | 0 | 0 | 0 | 0 | 0 | 2 |
| Frostbite | 65 | 4335 | **272** | **275** | **366** | 277 | 294 | 260 | 3 |
| Gopher | 258 | 2412 | **4752** | 4244 | **5307** | 4453 | 2690 | 4457 | 2 |
| Hero | 1027 | 30826 | **6777** | 3712 | **11434** | 9632 | 7913 | 6441 | 2 |
| Jamesbond | 29 | 303 | **544** | 470 | **408** | **418** | 383 | 347 | 4 |
| Kangaroo | 52 | 3035 | 1322 | **3300** | 3512 | 2418 | 3141 | **4218** | 4 |
| Krull | 1598 | 2666 | **7111** | **7060** | 6522 | 6927 | **10410** | 9715 | 2 |
| KungFuMaster | 258 | 22736 | 11055 | **22863** | 20046 | 20489 | 23036 | **24988** | 3 |
| MsPacman | 307 | 6952 | 1720 | **1822** | 1490 | 1557 | **2464** | **2401** | 2 |
| Pong | -21 | 15 | 8 | **13** | 18 | 18 | 14 | **21** | 3 |
| PrivateEye | 25 | 69571 | **100** | 88 | 100 | 216 | **364** | 85 | 3 |
| Qbert | 164 | 13455 | **3652** | 2974 | 2910 | 3220 | 4256 | **4546** | 3 |
| RoadRunner | 12 | 7845 | 15075 | **18014** | 14841 | 12841 | 16853 | **20482** | 4 |
| Seaquest | 68 | 42055 | 566 | **653** | 557 | 495 | 615 | **712** | 3 |
| UpNDown | 533 | 11693 | 5009 | **7486** | 6128 | 5098 | 4717 | **6623** | 3 |
| HNS mean | 0% | 100% | 106.6% | **119.4%** | 114.2% | 107.2% | 124.6% | **134.8%** | |
| HNS median | 0% | 100% | 33.2% | **42.6%** | **42.5%** | 35.5% | 35.0% | **43.8%** | |
| Obj-detectable | | | 144.0% | **160.8%** | 147.7% | 142.4% | 175.9% | **186.2%** | |
| Otherwise | | | **69.2%** | **78.0%** | **80.8%** | 72.0% | 73.3% | **83.4%** | |

## 4.1 ATARI 100K

We evaluate our method under the standard Atari 100k protocol (Kaiser et al., 2020), which limits training to $100,000$ environment frames. For each game, we report average performance across 5 random seeds, with each seed evaluated over 20 episodes. Performance is measured using human-normalized scores (HNS): HNS = (agent − random)/(human − random). To isolate the impact of OC representations, we focus here on ablations within our framework. External benchmarks against other MBRL approaches are presented in Appendix C.

We conduct experiments with two typical world modeling backbones: DreamerV3 (Hafner et al., 2023) and STORM (Zhang et al., 2023); two object extraction vision models: SAM2 (Hafner et al., 2023) and Cutie (Cheng et al., 2023); and two object representation methods: vector-based and mask-based. The mask-based representation follows the idea proposed in FOCUS (Ferraro et al., 2023); in our implementation, it is realized by feeding Cutie's segmentation masks into STORM. The results are shown in Table 1. Overall, OC methods clearly outperform the baseline. Cutie-based variants achieve higher scores than SAM2-based ones, while the mask-based representations underperform vector-based ones and are only on par with baselines. **We therefore select the best performing variant, Cutie-OC-STORM, in terms of HNS mean and median as our proposed algorithm, OC-STORM.**

To further examine the effectiveness of our method, we categorize the 26 games into two groups: those where all decision-relevant objects can be consistently identified by SAM2 or Cutie, and those where detection is incomplete.[*] For the second group, we further discuss the limitations of current vision models in Section 7. The results, reported at the bottom of Table 1, show that vector-based OC methods substantially outperform the baseline in the first category, while in the second category they still achieve performance comparable to the baseline. These findings highlight both the benefits of OC learning and the robustness of our framework.

**Interpretation of the results**   Although SAM2 achieves stronger results than Cutie on video segmentation benchmarks, our OC agents benefit more from Cutie's features. This discrepancy arises from their object-feature design: Cutie aggregates visual features within masked regions, whereas SAM2 produces a prototype vector intended for classification. Consequently, SAM2 features may lose positional and global context, providing weaker guidance for policy learning.

FOCUS-like mask-based methods mainly suffer from resolution issues: low-resolution inputs (e.g., $64 \times 64$ in world models) discard crucial object details, while high-resolution masks incur quadratic memory and computation costs. Moreover, raw masks are noisy, and thus their dynamics are difficult to predict. In contrast, the vector representation is semantically summarized from high-resolution input, which is more consistent and computationally efficient. Additional training curves comparing different representations are provided in Figure 4a.

## 4.2   HOLLOW KNIGHT

While the Atari benchmark is widely used in the RL community, it is not sufficient to verify the robustness of our proposed method. Atari's visual simplicity, where object pixels can often be separated from the background by well-defined pixel-value boundaries, allows methods like DreamerV3 and STORM to capture the environment dynamics almost perfectly in raw visual space. Such simplicity would be rarely seen in real-world scenarios. In contrast, the boss fights in Hollow Knight offer a more suitable testbed, where the visuals are much more complex, including many dynamic and distracting elements [†].

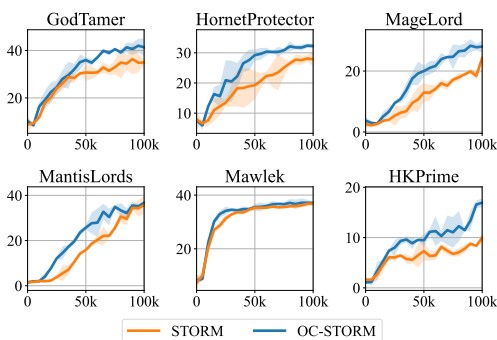

Figure 2: The training episode returns on Hollow Knight. We use a solid line to represent the mean of 3 seeds and use a semi-transparent background to represent the standard deviation.

For Hollow Knight, we similarly limit the number of samples to 100k, equivalent to approximately 3.1 hours of real-time gameplay at 9Hz. For each boss, we conduct three experiments with different random seeds.

---

[*]The first category ("Obj-detectable") includes Asterix, BankHeist, BattleZone, Boxing, ChopperCommand, DemonAttack, Jamesbond, Kangaroo, Krull, Pong, RoadRunner, Seaquest, and UpNDown. The second category includes Alien, Amidar, Assault, Breakout, CrazyClimber, Freeway, Frostbite, Gopher, Hero, KungFuMaster, MsPacman, PrivateEye, and Qbert.

[†]Please refer to the evaluation videos in the supplementary materials.

As seen in Figure 2, though the original STORM can also learn a good policy, our proposed OC method converges significantly faster and yields stronger performance in most cases, especially when the environment is more challenging, such as for Mage Lord and Pure Vessel. Numerical results are shown in Appendix E.1.

Since Hollow Knight is not yet an established benchmark, existing methods differ significantly in sample step limits, environment wrapping, reward functions, boss selection, among others. This makes direct comparisons with existing methods challenging. As the primary goal of this work is to improve MBRL through the use of OC representations, we therefore compare our results with the equivalent baseline algorithm STORM. Nevertheless, we include the results from Yang (2023) on the boss Hornet Protector for a rough comparison in Appendix E.6.

## 5 ANALYSIS

### 5.1 COMPLETENESS OF THE OBJECT REPRESENTATION

We utilize the output feature of the object transformer in the video model. While this feature theoretically contains all the state and positional information of an object, it is uncertain whether it fully captures these details in practice. Specifically, we need to determine if the masked pooling could potentially obscure positional information. The agent's performance, as demonstrated in Section 4, provides general quantitative evidence. Here, we present qualitative evidence to support this claim.

We trained a 4-layer ConvTranspose2d (Zeiler et al., 2010) decoder on the Atari Boxing game. It takes two vector-based object features as inputs, corresponding to the white and black players, respectively, to reconstruct the observation. The dataset was collected using a random policy, with $10,000$ frames for training and $1,000$ frames for validation. Sample reconstructions result from the validation set are shown in Figure 3a. This indicates that these features effectively capture the state and position of the objects. Both SAM2 and Cutie features can provide similar outcomes in this test, but the model takes a larger number of update steps to converge when using the SAM2 feature compared using the Cutie feature.

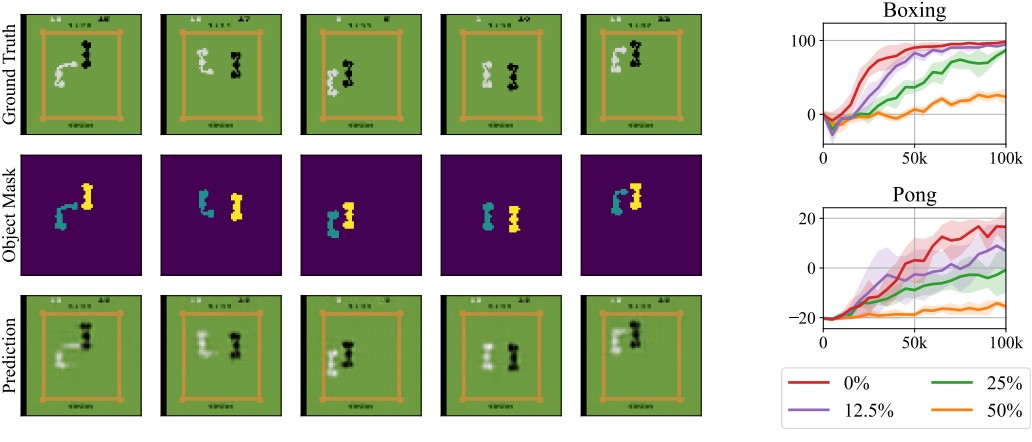

(a) Reconstructions with object features.     (b) Robustness analysis.

Figure 3: Analysis of the object feature. (a) Observation reconstructions on Atari Boxing with two object feature vectors as inputs. The object mask row highlights the relevant objects. (see Section 5.1 for details). (b) Training episode returns for Atari Boxing and Pong with $4$ different zeroing probabilities (see Section 5.2).

### 5.2 ANALYSIS OF SEGMENTATION MODEL FAILURES

To mimic segmentation model failures, we randomly set the object feature vector to $0$. This operation is identical to how we process the feature when the video model detects nothing, as described in

Appendix G.2. We conduct experiments on Atari Boxing and Pong with different zeroing probabilities. To avoid interference from visual inputs, we only use the object module in these experiments.

The results are presented in Figure 3b. As the detection accuracy of the vision model increases, the agent's performance improves accordingly. This also demonstrates the robustness of OC-STORM in handling unstable detection results. Additionally, since the zeroing process is purely random and the agent is trained only after the termination of each episode, every new episode during training serves as an indicator of test-time failure performance.

## 5.3 ADDITIONAL EXPERIMENTS WITH CONTINUOUS CONTROL

To evaluate the potential of OC-STORM on continuous control tasks, we conduct four experiments on the Meta-world (Yu et al., 2019) benchmark. We compare our results with MWM (Seo et al., 2022), which is also designed to help the world model focus on small dynamic objects. We choose one easy, two medium, and one hard tasks according to the MWM paper (see Seo et al. (2022) Appendix F, Experiments Details). These tasks are randomly selected to cover different objects and policies.

As shown in Figure 4b, OC-STORM generally exhibits higher sample efficiency than STORM. In some tasks, it also outperforms MWM in terms of efficiency and performance. This provides evidence that this approach can also perform well on continuous tasks out-of-the-box, without significant adaptation of the pipeline or extensive tuning for these very different continuous control environments.

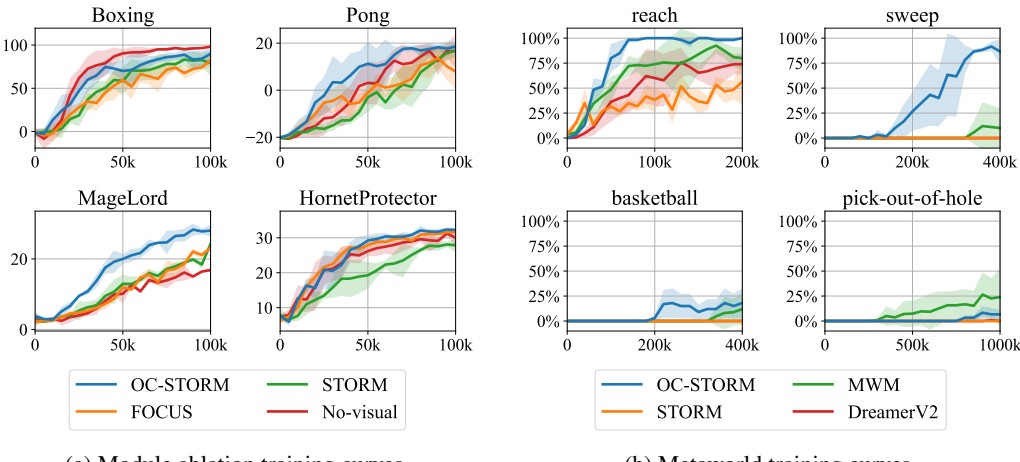

(a) Module ablation training curves.    (b) Metaworld training curves.

Figure 4: (a) Training episode returns under different input module configurations. STORM utilizes only the visual module, "No-visual" relies solely on the object module, and OC-STORM incorporates both, as described in Appendix A. FOCUS employs mask-based representations. The top two tasks are from Atari, while the bottom two are from Hollow Knight. See numerical results in Section 4.1 (b) Training success rates on 4 Meta-world tasks. The results of MWM (Seo et al., 2022) and DreamerV2 (Hafner et al., 2021) are from the MWM paper. See details in Section 5.3.

## 5.4 EXTRA ANALYSIS

We provide additional ablations in Appendix F, examining the policy design choices, the number of annotations, and the computational overhead of introducing pretrained vision models.

## 6 RELATED WORK

**Model-based RL**    Ha & Schmidhuber (2018) first demonstrated the feasibility of learning by imagination in pixel-based environments. SimPLe (Kaiser et al., 2020) further extended this idea to Atari games (Bellemare et al., 2013), though with limited efficiency. The Dreamer series (Hafner et al.,

2019; 2021; 2023) employs categorical VAEs and RNNs, to achieve robust performance across diverse domains. Dreamer introduces both a stable discretization method and a set of techniques for robust optimization across domains with diverse observations, dynamics, rewards, and goals. TWM (Robine et al., 2023) and STORM (Zhang et al., 2023) replace the RNN sequence model in Dreamer with transformers, enhancing parallelism during training. TWM encodes the observation, reward, and termination as three input tokens for the transformer, while STORM encodes them as a single token, demonstrating better efficiency. IRIS (Micheli et al., 2023), and improved efficiency variants $\Delta$-IRIS (Micheli et al., 2024) and REM (Cohen et al., 2024), utilize VQ-VAE (van den Oord et al., 2017) for multi-token latent representations. DIAMOND (Alonso et al., 2024) employs a diffusion process as the world model, further improving the final performance. All of these methods predominantly use an $L_2$ reconstruction loss for self-supervised learning.

**Object-centric RL**   Many attempts have been made to introduce OC learning to RL systems. However, to our knowledge, no existing methods have been directly applied to Atari games or Hollow Knight without leveraging internal game states or an extensive number of annotations. These OC learning methods broadly follow two main trends: two-stage and end-to-end.

**Two-stage**   methods usually first employ computer vision techniques to detect objects, then train the policy based on this object-level information. Current approaches often require labour-heavy task-specific fine-tuning (Devin et al., 2018; Liu et al., 2021), access to game memories (Delfosse et al., 2023; Jain, 2024), or leverage game-specific observation structures (Stanic et al., 2024). FOCUS (Ferraro et al., 2023), the most similar work to ours, is a model-based method that uses TrackingAnything (Yang et al., 2023) to generate segmentation masks, which are then fed into DreamerV2 (Hafner et al., 2021) for policy training. Using binary masks for object representation limits efficiency, which was discussed in Section 4. Moreover, FOCUS has only been tested on six clean-background fixed-camera robotics manipulation tasks, and hasn't been fully explored in more visually complex environments.

**End-to-end**   methods jointly learn the object perception and policy, often using unsupervised slot-based approaches (Locatello et al., 2020) to discover and represent objects. While these methods allow the visual module to be trained alongside the world model or policy network, their unsupervised learning nature leads to poor object detection quality, especially in noisy, real-world scenes. As a result, they are typically limited to visually simple OC benchmarks (Watters et al., 2024; Ahmed et al., 2021) and struggle to be adapted to visually complex tasks. Several model-based (Veerapaneni et al., 2019; Lin et al., 2020; van Bergen & Lanillos, 2022) and model-free (Yoon et al., 2023; Haramati et al., 2024) algorithms have used these ideas. Nakano et al. (2024) added slot attention to STORM, achieving stronger performance on the OCRL benchmark (Yoon et al., 2023).

## 7   LIMITATIONS

Our method has two main limitations, each of which corresponds to a potential future direction.

**Duplicated instances** Current video object segmentation algorithms are primarily developed and trained to track a single object. When a scene contains two or more identical objects, approaches like SAM2 or Cutie may fail to segment each object correctly and thus may affect performance.

**Geometric map representation** Our object representations are not well-suited for encoding geometric structures such as walls, boundaries, and navigable spaces. This limitation necessitates retaining raw visual inputs in our pipeline and reflects a broader challenge for current OC approaches in the visual domain.

These limitations are further illustrated and explained in Appendix I.

## 8   CONCLUSIONS

In this work, we introduced OC-STORM, an MBRL pipeline designed to improve sample efficiency in visually complex environments. By integrating recent advances in object segmentation and detection, we mitigate the limitations of traditional reconstruction-based MBRL methods, which may be

dominated by large background areas and overlook decision-relevant details. Through experiments across various domains and model configurations, we demonstrated that OC learning could be successfully implemented without relying on internal game states or extensive labelling, highlighting the adaptability of our method to complex, visually rich environments. OC-STORM represents a meaningful step toward combining modern computer vision with RL, offering an efficient framework for training agents in visually complex settings.

## ACKNOWLEDGEMENTS

The work of W.Z. and G.W. in this paper was supported by the National Natural Science Foundation of China under Grant U23B2059. The work of A.S. is part of a project of the European Union's Horizon Europe research and innovation programme under Grant Agreement No. 101120726, funded by UK Research and Innovation (UKRI) under the UK government's Horizon Europe funding guarantee 10085198.

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

## A   DETAILED MODEL STRUCTURE

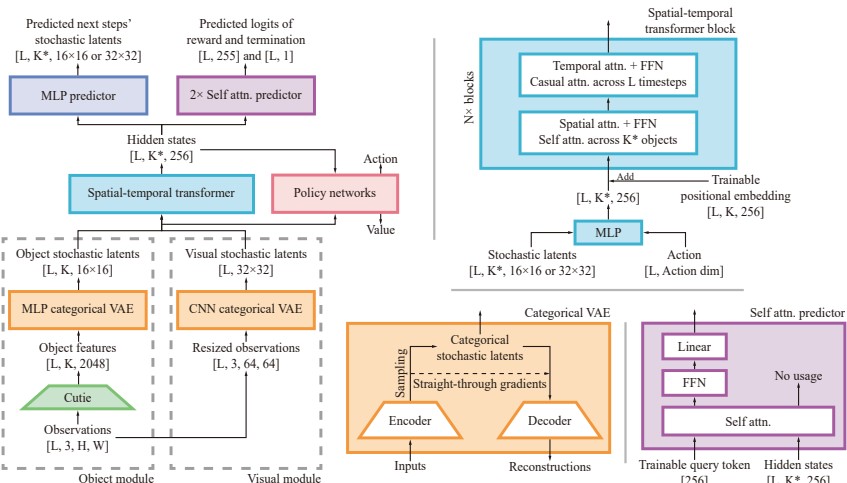

Figure 5: The model structure of our proposed OC-STORM. The tuples in square brackets represent the shapes of the corresponding tensors, where $L$ denotes the batch length or sequence length, $K$ is the number of objects, and $H$ and $W$ are the image height and width, respectively. The object module constitutes the proposed object-centric component, while the visual module processes resized raw observations. $K^*$ equals $K$ when only object module is used, equals 1 when only visual module is used, and equals $K + 1$ when both modules are used. The trainable token and positional embeddings are broadcast to match the shapes of the corresponding tensors. The reward logit is 255-dimensional and used for the symlog two-hot loss (Hafner et al., 2023).

## B   LOSS FUNCTIONS

### B.1   WORLD MODEL LEARNING

The world model is trained in a self-supervised manner, optimizing it end-to-end. Our setup closely follows DreamerV3 (Hafner et al., 2023), with details presented for completeness. We use mean squared error (MSE) loss for reconstructing the original inputs, symlog two-hot loss $\mathcal{L}_{\text{sym}}$ (Hafner et al., 2023) for reward prediction, and binary cross-entropy (BCE) loss for termination signal prediction. These losses, collectively referred to as prediction loss, are defined as:

$$\mathcal{L}_{\text{pred}}(\phi) = \underbrace{||\hat{s}_t - s_t||_2^2}_{\text{Reconstruction loss}} + \underbrace{\mathcal{L}_{\text{sym}}(\hat{r}_t, r_t)}_{\text{Reward loss}} + \underbrace{\tau_t \log \hat{\tau}_t + (1 - \tau_t) \log(1 - \hat{\tau}_t)}_{\text{Termination loss}}. \tag{5}$$

The dynamics loss $\mathcal{L}_t^{\text{dyn}}(\phi)$ guides the sequence model in predicting the next distribution. The representation loss $\mathcal{L}_t^{\text{rep}}(\phi)$ allows the encoder's output to be weakly influenced by the sequence model's prediction, ensuring that the dynamics are not overly difficult to learn. These losses are identical Kullback–Leibler (KL) divergence losses except for their gradient propagation settings. We use $\text{sg}(\cdot)$ to denote the stop gradient operation. The dynamics and representation losses are defined as:

$$\mathcal{L}_{\text{dyn}}(\phi) = \max\left(1, \text{KL}\left[\text{sg}(q_\phi(z_{t+1}|s_{t+1})) \,||\, g_\phi^{\text{dyn}}(\hat{z}_{t+1}|h_t)\right]\right), \tag{6a}$$

$$\mathcal{L}_{\text{rep}}(\phi) = \max\left(1, \text{KL}\left[q_\phi(z_{t+1}|s_{t+1}) \,||\, \text{sg}(g_\phi^{\text{dyn}}(\hat{z}_{t+1}|h_t))\right]\right). \tag{6b}$$

The max operation represents free bits for KL divergence, encouraging the model to focus on optimizing prediction losses for better feature extraction if the KL divergence is too small.

The total loss function for training the world model is calculated as follows, where $\mathbb{E}_{\mathcal{D}}$ denotes the expectation over samples from the replay buffer:

$$\mathcal{L}(\phi) = \mathbb{E}_{\mathcal{D}}\left[\mathcal{L}_{\text{pred}}(\phi) + \mathcal{L}_{\text{dyn}}(\phi) + 0.5\mathcal{L}_{\text{rep}}(\phi)\right]. \tag{7}$$

The coefficient of 0.5 for $\mathcal{L}_{\text{rep}}$ is used to prevent posterior collapse (Lucas et al., 2019), a situation where the model produces the same distribution for different inputs, causing the dynamics loss to trivially converge to 0. The imbalanced KL divergence loss helps to mitigate this issue.

## B.2 POLICY LEARNING

The policy learning approach closely follows that of DreamerV3 (Hafner et al., 2023), with modifications specific to our method. The key differences lie in the input for the policy and the action dimension for the game Hollow Knight. We use the concatenation of object latents, object hidden states, visual latents, and visual hidden states as input features. For Hollow Knight, a multi-discrete action space is employed.

The agent learns entirely on the imagined trajectories generated by the world model. To begin the imagination process, we first sample a short contextual trajectory from the replay buffer. During imagination, future environmental inputs $s_{t+1:L}$ are unknown, and sampling from the posterior distribution $q_\phi(z_t|s_t)$ is unavailable. Thus, we sample the latent variable from the prior distribution $g_\phi^{\text{dyn}}(\hat{z}_{t+1}|h_t)$ and optimize the policy over $\hat{z}_{t+1}$. However, during testing, the agent interacts directly with the environment, allowing access to the posterior distribution of the last observation. This introduces a difference in notation. For simplicity, we do not distinguish between $z_t$ and $\hat{z}_t$ in the following descriptions.

The agent uses both the latent variable $\hat{z}_t$ and hidden states $h_t$ as inputs, as defined below:

$$\text{Critic:} \quad V_\psi(z_t, h_t) \approx \mathbb{E}_{\pi_\theta, \phi}\left[\sum_{k=0}^{T} \gamma^k r_{t+k}\right],$$

$$\text{Actor:} \quad a_t \sim \pi_\theta(a_t|z_t, h_t). \tag{8}$$

Here, $T$ is the number of timesteps in the episode. We use two separate MLPs for the critic and actor networks. The symbol $\phi$ indicates that the trajectories are generated within the imagination process of the world model.

For value loss, we employ the $\lambda$-return $G_t^\lambda$ (Sutton & Barto, 2018; Hafner et al., 2023) to improve value estimation. It is recursively defined as follows, where $\hat{r}_t$ is the reward predicted by the world model, and $\hat{\tau}_t$ represents the predicted termination signal:

$$G_t^\lambda \doteq \hat{r}_t + \gamma(1 - \hat{\tau}_t)\left[(1-\lambda)V_\psi(z_{t+1}, h_{t+1}) + \lambda G_{t+1}^\lambda\right], \tag{9a}$$

$$G_L^\lambda \doteq V_\psi(z_L, h_L). \tag{9b}$$

To regularize the value function, we maintain an exponential moving average (EMA) of the critic's parameters, as defined in Equation equation 10. This regularization technique stabilizes training and helps prevent overfitting, where $\psi_t$ represents the current critic parameters, $\sigma$ is the decay rate, and $\psi_{t+1}^{\text{EMA}}$ denotes the updated critic parameters:

$$\psi_{t+1}^{\text{EMA}} = \sigma\psi_t^{\text{EMA}} + (1-\sigma)\psi_t. \tag{10}$$

For policy gradient loss, we apply return-based normalization for the advantage value. The normalization ratio $S$ is defined in Equation equation 11 as the range between the 95th and 5th percentiles of the $\lambda$-return $G_t^\lambda$ across the batch (Hafner et al., 2023):

$$S = \text{percentile}(G_t^\lambda, 95) - \text{percentile}(G_t^\lambda, 5). \tag{11}$$

The complete loss functions for the actor-critic algorithm are given by Equation equation 12:

$$\mathcal{L}(\theta) = \mathbb{E}_{\pi_\theta, \phi}\left[-\text{sg}\left(\frac{G_t^\lambda - V_\psi(s_t)}{\max(1, S)}\right)\ln \pi_\theta(a_t|z_t, h_t) - \eta H\big(\pi_\theta(a_t|z_t, h_t)\big)\right], \tag{12a}$$

$$\mathcal{L}(\psi) = \mathbb{E}_{\pi_\theta, \phi}\left[\mathcal{L}_{\text{sym}}\Big(V_\psi(z_t, h_t), \text{sg}(G_t^\lambda)\Big) + \mathcal{L}_{\text{sym}}\Big(V_\psi(z_t, h_t), \text{sg}\big(V_{\psi^{\text{EMA}}}(z_t, h_t)\big)\Big)\right]. \tag{12b}$$

Here, $H(\cdot)$ denotes the entropy of the policy distribution, and $\eta = 1 \times 10^{-3}$ is the coefficient for entropy loss.

# C   COMPARISON WITH OTHER MBRL ALGORITHMS

Table 2: Game scores, overall human-normalized scores, and computing time on the Atari 100k benchmark. Bold indicates scores within 5% of best. The last column '#obj' denotes manually annotated objects per game. The DreamerV3 scores here are the original data from their paper (Hafner et al., 2023). SPR (Schwarzer et al., 2021) is a typical sample-efficient model-free method. IRIS (Micheli et al., 2023) and TWM (Robine et al., 2023) are transformer-based MBRL approaches, and Δ-IRIS (Micheli et al., 2024) is an upgraded version. DIAMOND (Alonso et al., 2024) is a diffusion-based MBRL approach.

| Game | Random | Human | SPR | IRIS | TWM | DreamerV3 | Δ-IRIS | DIAMOND | OC-STORM |
|------|-------|-------|-----|------|-----|-----------|--------|---------|----------|
| Alien | 228 | 7128 | 842 | 420 | 675 | **1278** | 599 | 744 | 1101 |
| Amidar | 6 | 1720 | 180 | 143 | 122 | 120 | 51 | **226** | 163 |
| Assault | 222 | 742 | 566 | **1524** | 683 | 741 | 1435 | **1526** | 1270 |
| Asterix | 210 | 8503 | 962 | 854 | 1117 | 1020 | 2001 | **3698** | 1754 |
| BankHeist | 14 | 753 | 345 | 53 | 467 | 422 | **1206** | 20 | 1075 |
| BattleZone | 2360 | 37188 | 14834 | 13074 | 5068 | **20800** | 10365 | 4702 | 4590 |
| Boxing | 0 | 12 | 36 | 70 | 78 | 87 | 56 | 87 | **92** |
| Breakout | 2 | 30 | 20 | 84 | 20 | 11 | **226** | 132 | 52 |
| ChopperCommand | 811 | 7388 | 946 | 1565 | 1697 | **2440** | 1101 | 1370 | 2090 |
| CrazyClimber | 10780 | 35829 | 36700 | 59324 | 71820 | 80060 | 70920 | **99168** | 84111 |
| DemonAttack | 152 | 1971 | 518 | **2034** | 350 | 454 | 884 | 288 | 411 |
| Freeway | 0 | 30 | 19 | 31 | 24 | 0 | 31 | 33 | 0 |
| Frostbite | 65 | 4335 | 1171 | 259 | 1476 | **3914** | 287 | 274 | 260 |
| Gopher | 258 | 2412 | 661 | 2236 | 1675 | 2252 | **9349** | 5898 | 4457 |
| Hero | 1027 | 30826 | 5859 | 7037 | 7254 | **13324** | 6235 | 5622 | 6441 |
| Jamesbond | 29 | 303 | 366 | 463 | 362 | **490** | 345 | 427 | 347 |
| Kangaroo | 52 | 3035 | 3617 | 838 | 1240 | 2840 | 1573 | **5382** | 4218 |
| Krull | 1598 | 2666 | 3682 | 6616 | 6349 | 8604 | 6392 | 8610 | **9715** |
| KungFuMaster | 258 | 22736 | 14783 | 21760 | **24555** | 25560 | 25159 | 18714 | **24988** |
| MsPacman | 307 | 6952 | 1318 | 999 | 1588 | 1400 | 1175 | 1958 | **2401** |
| Pong | -21 | 15 | -5 | 15 | 19 | -5 | 15 | **20** | **21** |
| PrivateEye | 25 | 69571 | 86 | 100 | 87 | **3238** | 100 | 114 | 85 |
| Qbert | 164 | 13455 | 866 | 746 | 3331 | 3700 | 3438 | **4499** | 4546 |
| RoadRunner | 12 | 7845 | 12213 | 9615 | 9109 | 18440 | 10622 | **20673** | 20482 |
| Seaquest | 68 | 42055 | 558 | 661 | 774 | **964** | 895 | 551 | 712 |
| UpNDown | 533 | 11693 | 10859 | 3546 | 15982 | **49456** | 8091 | 3856 | 6623 |
| HNS mean | 0% | 100% | 61.6% | 104.6% | 95.6% | 128.2% | 138.3% | **145.9%** | 134.8% |
| HNS median | 0% | 100% | 39.6% | 28.9% | 50.5% | 48.7% | **59.4%** | 37.3% | 43.8% |
| Average compute | | | PyTorch | PyTorch | PyTorch | Jax | PyTorch | PyTorch | PyTorch |
| 4090 GPU days | | | 0.1 | 3.5 | 0.4 | 0.1 | 1.0 | 2.9 | 0.2 |

# D    REVIEW OF OBJECT REPRESENTATION METHODS

Table 3: A brief review of state-of-the-art methods in different fields of computer vision related to object-centric reinforcement learning.

| Method | Description |
|---|---|
| Cutie | Retrieval-based semi-supervised video object segmentation algorithm. Provides compact vector representation of objects using an object transformer (Cheng et al., 2023). |
| XMem | Retrieval-based semi-supervised video object segmentation algorithm with multi-level memory system. Lacks compact object representation (Cheng & Schwing, 2022). |
| SAM | Open-set image segmentation algorithm. Generates masks via user prompts but requires a prompt for each frame in a video (Kirillov et al., 2023). |
| TrackAnything | SAM + XMem for fine-grained video segmentation. Requires twice the computational resources compared to XMem (Yang et al., 2023). |
| PerSAM | One-shot enhancement of SAM. Difficult to expand to few-shot cases (Zhang et al., 2024). |
| SAM2 | State-of-the-art multi-usage video object segmentation algorithm. Capable of providing compact vector representations. (Ravi et al., 2024). |
| YOLO | Closed-set object detection algorithm. Requires extensive annotations for training or fine-tuning (Redmon et al., 2016; Jocher et al., 2023). |
| GroundingDINO | Open-set object detection algorithm. Generates object bounding boxes using natural language prompts but struggles with rare or abstract cases, such as video game players (Liu et al., 2023). |
| Slot attention | Unsupervised image object discovery and segmentation algorithm. Underperforms compared to supervised methods (Locatello et al., 2020). |
| SAVi | Unsupervised video object discovery and segmentation algorithm. Evaluation is semi-supervised, though training is unsupervised. Underperforms compared to semi-supervised methods (Kipf et al., 2022; Elsayed et al., 2022). |
| Omnimotion | Video point-tracking algorithm. Provides pixel-level tracking but lacks object-level information and does not support few-shot scenarios (Wang et al., 2023). |
| Unimatch | Dense optical flow and depth estimation algorithm. Useful for moving identity detection but cannot extract object-level information and struggles with out-of-domain generalization (Xu et al., 2023). |

# E    HOLLOW KNIGHT

## E.1    NUMERICAL RESULTS

Table 4: Episode returns and win rates (WR) of STORM and the proposed OC-STORM on Hollow Knight. Each seed's performance is measured by the mean episode return across 20 runs, and the average of these 3 mean returns is reported. The "#obj" column shows the number of annotated objects for a boss. Scores that are the highest or within 5% of the highest score are highlighted in bold. To evaluate the upper limit of our agent, we conduct a 400k run on Pure Vessel, which demonstrates that our agent can defeat one of the most difficult bosses in the game with sufficient training.

| Boss name | Random | Optimal | STORM | OC-STORM | STORM WR | OC-STORM WR | #obj |
|---|---|---|---|---|---|---|---|
| God Tamer | 10 | 56 | 35.0 | **41.7** | **70.0%** | 55.0% | 4 |
| Hornet Protector | 7 | 37 | 28.1 | **32.4** | 66.7% | **100.0%** | 2 |
| Mage Lord | 3 | 38 | 19.6 | **28.0** | 5.0% | **48.0%** | 3 |
| Mantis Lords | 2 | 42 | 33.2 | **35.2** | 71.7% | **83.3%** | 3 |
| Mawlek | 8 | 41 | **36.9** | **37.2** | **98.3%** | **98.3%** | 3 |
| Pure Vessel | 2 | 55 | 8.9 | **15.7** | 0.0% | 0.0% | 2 |
| Pure Vessel (400k) | 2 | 55 | 25.3 | **35.0** | 6.7% | **13.3%** | 2 |

### E.2 Related Work

Despite its popularity among players, Hollow Knight has seen limited use as a benchmark for research in reinforcement learning. We introduce repositories, published research, and other relevant resources that leverage or explore Hollow Knight as a benchmark. Cui (2021) employs DQN (Mnih et al., 2015) and its variants but requires modding the game background to black to enhance character perception. Yang (2023) uses the Rainbow algorithm (Hessel et al., 2018) with additional techniques like DrQ (Yarats et al., 2022; 2021), achieving high win rates against several of the game's bosses. Yang's repository has been widely forked and adopted. Building on his work, Lee (2023) studies the effect of reward shaping, while Sun (2024) focuses on improving training efficiency by tuning the game interaction configuration and switching to the PPO algorithm (Schulman et al., 2017). Jain (2024) leverages internal game states to extract hitboxes as input for the algorithm, representing them as segmentation masks that are passed to DQN or PPO.

### E.3 Environment Configuration

Hollow Knight is a modern video game developed with Unity (Technologies, 2005). To our knowledge, efficient simulators for this game, such as those available for Atari (Brockman et al., 2016; Towers et al., 2023), are not publicly available. Therefore, we developed a custom wrapper that captures screenshots of the game at 9 FPS and sends keyboard signals to execute actions. The 9 FPS rate is a choice based on the author's experience with the game and considerations for computational efficiency. To obtain reward signals, we developed a modding plugin (Bham & Wyza, 2017) that logs when the player-controlled character (the Knight) either hits an enemy or is hit. Our wrapper then parses this log file to generate reward and termination signals.

The game execution and agent training are conducted on a Windows machine. To monitor training progress and statistics without interrupting the game, we needed a method to send keyboard inputs to a background or unfocused window. However, Windows lacks an API for this purpose. As a result, the game window must remain in the foreground, fully occupying the training device and hindering monitoring. To address this, we utilized a Hyper-V (Cooley, 2022) Windows virtual machine to run the game in the background, with Ray (Moritz et al., 2018) facilitating communication between the host and virtual machine. Training and processing occur on the host machine, while the virtual machine handles interactions with the environment. This setup can be extended to distributed nodes, with some handling game rendering and others managing training tasks.

For in-game configuration, the charms (Wiki, 2018) are set to Unbreakable Strength, Quick Slash, Soul Catcher, and Shaman Stone across all experiments. This configuration is chosen to explore the agent's fighting potential rather than its glitch-finding abilities.

### E.4 Action Space

All previous works utilize a human-specified action space rather than the original keyboard inputs. For example, in the Yang (2023) implementation, short and long jumps are treated as two distinct actions, which are originally controlled by the duration of the jump button press. His environment wrapper handles this difference with a fixed command. While this design reduces the exploration and computational costs for reinforcement learning agents, it cannot capture the full range of possible actions in the game. Advanced operations, as demonstrated in this video (CrankyTemplar, 2018), require full control of the keyboard. Therefore, we design the action space as a multi-binary-discrete one that directly binds to the press and release of the physical keyboard, which will be explained in Section E.4.

We design the action space as a multi-binary-discrete space directly mapped to the press and release states of eight specific keys on the keyboard. These keys include W, A, S, and D for movement directions, and J, K, L, and I for attack, jump, dash, and spell actions, respectively. Each key's state is represented as a binary variable, where 0 corresponds to a key release and 1 corresponds to a key press. The action can therefore be described as $a \in \{0,1\}^8$, with each element representing the binary state of one key. The key's state is maintained between frames, and a toggle signal is sent only when there is a change in the key state from $0 \rightarrow 1$ (press) or $1 \rightarrow 0$ (release).

The probability of an action is determined by the independent probabilities of each key's state:

$$\pi_\theta(a|z,h) = \prod_{k=1}^{8} \pi_\theta(a^k|z,h) \tag{13}$$

where $a^k$ denotes the state of the $k$-th key.

The entropy of the action space, $H(\pi_\theta(a|z,h))$, is the sum of the entropies of the individual key states:

$$H(\pi_\theta(a|z,h)) = \sum_{k=1}^{8} H(\pi_\theta(a^k|z,h)) \tag{14}$$

This design provides fine-grained control over the agent's actions, allowing for the execution of complex manoeuvres while maintaining a tractable exploration space for reinforcement learning.

### E.5    REWARD SHAPING IN HOLLOW KNIGHT

Most existing methods (Yang, 2023; Sun, 2024; Jain, 2024; Cui, 2021; Lee, 2023) for Hollow Knight use a reward structure of +1 for hitting an enemy and -1 for taking damage. Some approaches modify the weighting ratios, while others introduce auxiliary rewards for performing specific actions. However, we found that these settings are suboptimal for training reinforcement learning agents.

Our method assigns a +1 reward signal for hitting an enemy and a virtual termination signal upon being hit. The game continues until the episode naturally ends. The termination signal is stored in the replay buffer for training the world model, treating health loss as a life-loss event. Leveraging life-loss information is a common technique that aids in value estimation (Ye et al., 2021; Micheli et al., 2023; Zhang et al., 2023; Alonso et al., 2024). Additionally, the Knight can damage enemies in multiple ways, and these damages are normalized against the base attack damage to compute the positive reward.

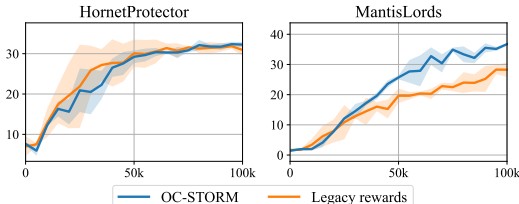

Figure 6: Training episode returns for Hollow Knight's Hornet Protector and Mantis Lords under different reward settings. "Legacy rewards" refer to the reward scheme used in prior works. For comparison, we aligned the returns from "legacy rewards" with our baseline settings by accounting for lost health.

Here, we present the key differences between the two reward settings. As illustrated in Figure 6, our reward configuration is more robust than those used in previous studies, resulting in significantly improved performance, especially in more challenging environments like Mantis Lords. This improvement can be analyzed from two perspectives:

1. Terminating the episode upon being hit better aligns with human cognition and the agent's expected behaviour. The aim is for the agent to deal as much damage as possible without taking any. While this may seem aggressive, raising concerns that the agent might sacrifice itself to deal more damage, neither our qualitative nor quantitative results show this tendency. Survival naturally offers more opportunities to deal with future damage, which the agent learns to prioritize. Although applying a negative penalty for being hit could prevent the agent from sustaining multiple consecutive hits in highly unfavourable situations, such scenarios should not occur under an optimal or near-optimal policy.

2. While maintaining the same optimal policy, truncating future rewards upon being hit significantly reduces the variance in value estimation. Hollow Knight is a highly stochastic

environment where bosses behave aggressively yet unpredictably. Estimating value directly over an episode (lasting approximately 300 to 700 timesteps) is inherently challenging in such settings.

### E.6 COMPARISON WITH A MODEL-FREE BASELINE

As introduced in Section E.2, Yang's repository (Yang, 2023) is a widely recognized implementation within the community. In this section, we compare the performance against the boss Hornet Protector.

Yang's reward structure assigns +0.8 for hitting an enemy and -0.8 for taking damage, with additional auxiliary rewards on the order of $1 \times 10^{-4}$ for various actions. A small feedback reward is also given at the end of each episode. The choice of a 0.8 weight factor for rewards reflects the use of +1/-1 reward clipping, with a margin reserved for the auxiliary rewards. We provide a broad comparison with this approach below.

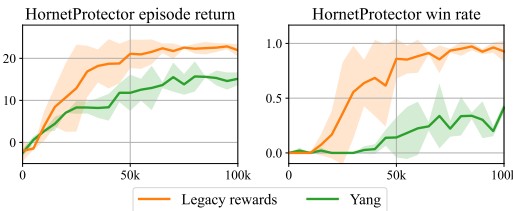

Figure 7: Training episode returns and win rates on Hollow Knight's Hornet Protector with our proposed method and Yang's (Yang, 2023) method. "Legacy rewards" are as described in Section E.5. We applied some preprocessing to align the two returns for easier comparison, so the "legacy rewards" curve may appear different from the one shown in the previous section. The win rate is more straightforward and can be used for comparison without changing.

As shown in Figure 7, our implementation is more efficient than Yang's. As we noted in Section 4.2, there are significant differences between our methods, making this **not strictly** a fair comparison from an algorithmic standpoint. This comparison is intended solely to demonstrate the efficiency of our implementation.

Additionally, Yang claims that his agent can achieve 10 wins out of 10 battles, which is accurate despite his win rate in our plot appearing to be lower than 100%. Two reasons may lead to this. First, his original sample steps are greater than ours, which may account for differences in performance. Second, our in-game charm configuration (Wiki, 2018) differs from the one used in his implementation. When testing Yang's implementation, we retained our current charm settings, which likely impacted the win rate results.

## F ADDITIONAL AND ANALYSIS

### F.1 ATTENTION-BASED POLICY OR MLP-BASED POLICY

When handling multiple objects, an attention-based policy network like the self-attention predictor described in Appendix A is a natural choice. A previous work OC-SA (Stanic et al., 2024) has also explored such structure. However, we still design our actor and critic networks as MLPs which take the concatenation of object latent variables and hidden states as input.

We found that the attention-based policy tends to overfit pre-learned behaviours and makes it hard to learn new knowledge. This will not be a major issue in stationary games like Boxing but will face trouble in non-stationary games like Pong. For example, the attention-based policy can quickly learn how to catch the ball but cannot efficiently learn how to score against the opponent. On the one hand, we can confirm this by visually checking the rendered episodes. On the other hand, numerically speaking, we can observe the episode length of playing Pong. If the episode length increases while the episode returns remain at the same level, then we can tell that the agent learns how to catch the ball, but is stuck in that local optimum.

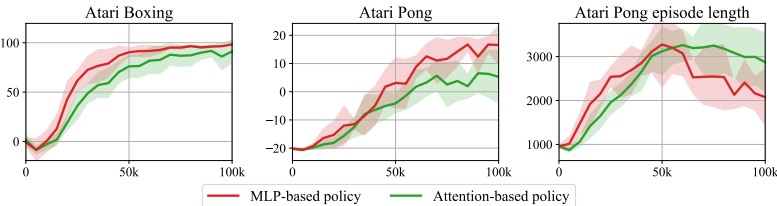

Figure 8: Training episode returns for Atari Boxing, Pong and episode lengths for Pong of attention-based policy and MLP-based policy. The attention-based policy can learn as quickly as the MLP-based policy for catching the ball but struggles to transition to the scoring phase in Atari Pong.

As the results plotted in Figure 8, we can tell that the attention-based policy suffers from that issue. The episode length of both policies rises at a similar speed before 50k steps, but it declines slower for the attention-based policy after that. The experiments are conducted using only the object module, so the MLP-based policy curves are identical to the "vector" ones in Figure 4a. As the visual latent itself contains all the information, the agent can choose only to use that part of the information and thus may affect our judgement on the effectiveness of the attention-based policy.

Though the attention-based policy has the potential to handle a dynamic number of objects, our experiments are conducted on a fixed number. As it doesn't demonstrate superior performance than the MLP-based policy in our case, we always use MLP-based in other tasks for consistency in evaluation.

### F.2 IMPACT OF THE NUMBER OF ANNOTATIONS

Since Cutie is a retrieval-based algorithm that stores past frames and masks in a buffer for reference, it naturally supports the use of multiple annotation masks beyond the first frame by substituting model-generated masks in the buffer. Incorporating more label masks can capture a wider range of object states, leading to more consistent segmentation results. However, reducing the number of labels can further lower annotation costs and computational complexity. In this section, we explore the impact of the number of labels on agent performance. To eliminate the influence of visual input, we conduct these experiments using only the object module.

As shown in Figure 9, increasing the number of annotation segmentation masks enhances the robustness of the agent's performance, even in visually static environments like Atari Boxing and Pong. In these environments, a single frame can include all necessary objects for decision-making. However, Cutie may lose track of objects if their states deviate significantly from those in the labelled masks, such as the punching state versus the standing state in Boxing, or the paddle in Pong when it is partially off-screen and appears shorter than when centred.

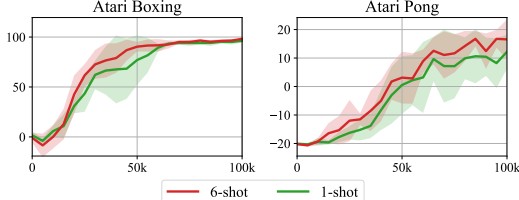

Figure 9: Training episode returns for Atari Boxing and Pong with different numbers of annotation segmentation masks. Increasing the number of annotation masks enhances the robustness of the agent's performance.

Moreover, in complex environments like Hollow Knight and Minecraft, a single frame may not capture all objects, which often necessitates additional segmentation masks. For consistency in evaluation, we use six annotation segmentation masks for Atari and twelve for Hollow Knight.

### F.3 THE COMPUTATIONAL OVERHEAD OF OC-STORM

The computational overhead of OC-STORM on Atari games with an NVIDIA GeForce RTX 3090 is shown in Table 5. The input resolution for Cutie is $420 \times 320$ (double the original $210 \times 160$). Thus the computational cost of introducing Cutie is acceptable in many cases.

Table 5: The computational overhead of OC-STORM on Atari games. The three numbers in a block here mean sample or evaluation speed (iterations/second), training speed (iterations/second) and hours spent for a train with a 100k sample budget, respectively.

| Algorithm | 0 objects | | | 1 object | | | 2 objects | | | 3 objects | | |
|---|---|---|---|---|---|---|---|---|---|---|---|---|
| STORM (visual module only) | 114 it/s | 8.1 it/s | 3.67 h | - | | | - | | | - | | |
| OC-STORM (obj module only) | - | | | 32 it/s | 8.8 it/s | 4.02 h | 32 it/s | 8.5 it/s | 4.14 h | 31 it/s | 7.8 it/s | 4.46 h |
| OC-STORM (both modules) | - | | | 28 it/s | 5.9 it/s | 5.70 h | 27 it/s | 5.5 it/s | 6.08 h | 27 it/s | 5.3 it/s | 6.27 h |

We report the latency of each part in our pipeline in Table 6. As shown, the introduced overhead is modest, and the system maintains real-time performance throughout all evaluated settings.

Table 6: Average system-level runtime (ms) of the proposed model on a single NVIDIA RTX 4090 GPU, with segmentation inputs at a resolution of $420 \times 320$ pixels. The 10 objects case uses a manually constructed synthetic configuration for testing.

| Module | 2 objects | 3 objects | 4 objects | 10 objects |
|---|---|---|---|---|
| Cutie Load | | | 1500 | |
| **Sample step** | 15 | 15 | 16 | 21 |
| Segmentation Inference | 11 | 11 | 12 | 15 |
| Policy Execution | 4 | 4 | 4 | 6 |
| **Train step** | 180 | 196 | 212 | 332 |
| World Model Gradient Step | 40 | 41 | 42 | 47 |
| World Model Imagination | 113 | 127 | 142 | 257 |
| First Step Imagination | 4.5 | 5.3 | 5.5 | 11.0 |
| Last Step Imagination (16 steps) | 5.6 | 6.3 | 7.3 | 13.4 |
| Policy Gradient Step | 27 | 28 | 28 | 28 |

## G    DETAILS FOR THE USE OF SAM2 AND CUTIE

### G.1    NUMBER OF ANNOTATIONS, INPUT RESOLUTIONS, AND MODEL SIZE

To prompt Cutie, we use 6 annotation masks per Atari game and 12 per Hollow Knight boss. One potential critique is that few-shot annotation requires prior knowledge of the environment, which may seem unsuitable for general agent learning. However, we view this process as akin to informing the agent of certain task rules. While rewards can reflect task rules, they are often too sparse to facilitate an understanding of complex environments. Just as humans may initially struggle to understand how to play a game without being told the rules, there is no reason not to inform agents of key objects. Therefore, we believe this pipeline holds practical value in many cases.

For **Atari**, we upscale the observation from $210 \times 160$ to $420 \times 320$. This upscaling aids in the identification of small objects in Atari games, such as the ball in Pong and Breakout. For each game, we hand annotate 6 masks. For **Hollow Knight**, we resize the observation's shorter side to 480p while maintaining the aspect ratio before inputting it into the Cutie. For each game, we hand annotate 12 masks.

For Cutie, we use `cutie-small-mega.pth`; for SAM2, we use `sam2.1_hiera_small.pt`, along with the corresponding configurations, respectively.

### G.2    MODIFICATIONS FOR INTEGRATION WITH STORM

We make no modifications to the official implementation, except for caching and copying internal variables. The only special process involves setting the object feature vector to 0 when the model loses track of the object. This can prevent the world model from fitting random dynamics in such cases.

For **SAM2**, this is rather straightforward since the model naturally contains a model to predict whether the target is occluded, and we can directly use this score as the determinant.

For **Cutie**, the model uses an attention guidance mask within its object transformer, which restricts which visual features the object feature can attend to. This mask is trained as part of an auxiliary segmentation task. When the attention guidance mask is set to all 1s (0 allows attention and 1 rejects it), indicating that Cutie cannot find strong evidence of the object's presence in the scene, the transformer theoretically should reject all attention from the visual features.

However, in this situation, Cutie inverts the mask, allowing the object feature to attend to all visual features in an attempt to search for the object in the scene. As a result, the attention becomes scattered across the observation space, leading to unpredictable output for the object feature. Thus, we set the object feature vector to 0 when the attention guidance mask is entirely 1.

# H HYPERPARAMETERS

Table 7: Hyperparameters for both Atari and Hollow Knight. The life loss information configuration aligns with the setup used in EfficientZero (Ye et al., 2021). Regarding data sampling, each time we sample $B_1$ trajectories of length $T$ for world model training, and sample $B_2$ trajectories of length $C$ for starting the imagination process. The train ratio is defined as the number of gradient steps over the number of environment steps.

| Hyperparameter | Symbol | Value |
|---|---|---|
| Transformer layers | $K$ | 2 |
| Transformer feature dimension | $D$ | 256 |
| Transformer heads | - | 4 |
| Dropout probability | $p$ | 0.1 |
| World model training batch size | $B_1$ | 32 |
| World model training batch length | $T$ | 32 |
| Imagination batch size | $B_2$ | 512 |
| Imagination context length | $C$ | 4 |
| Imagination horizon | $L$ | 16 |
| Train ratio | - | 1 |
| Environment context length | - | 16 |
| Gamma | $\gamma$ | 0.985 |
| Lambda | $\lambda$ | 0.95 |
| Entropy coefficiency | $\eta$ | $1 \times 10^{-3}$ |
| Critic EMA decay | $\sigma$ | 0.98 |
| Optimizer | - | Adam (Kingma & Ba, 2015) |
| Activation functions | - | SiLU (Elfwing et al., 2018) |
| World model learning rate | - | $1.0 \times 10^{-4}$ |
| World model gradient clipping | - | 1000 |
| Actor-critic learning rate | - | $3.0 \times 10^{-5}$ |
| Actor-critic gradient clipping | - | 100 |
| Gray scale input | - | False |
| Frame stacking | - | False |
| Atari frame skipping | - | 4 (max over last 2 frames) |
| Hollow Knight FPS | - | 9 |
| Use of life information | - | True |

## I  ILLUSTRATION OF LIMITATIONS

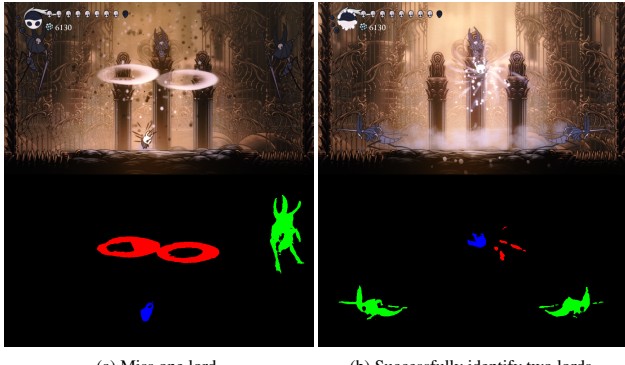

(a) Miss one lord.  (b) Successfully identify two lords.

Figure 10: Sample frame and segmentation masks generated by Cutie from the Hollow Knight Mantis Lords. Cutie may lose track of one of the lords (represented with green masks). This tracking issue is more likely to occur not only in this scenario but also in other environments where duplicated instances are present, compared to scenes with a single instance.

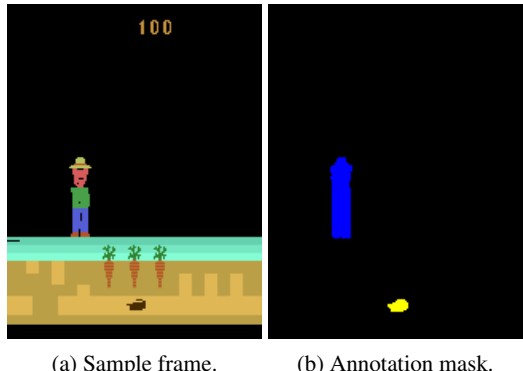

(a) Sample frame.  (b) Annotation mask.

Figure 11: Sample frame and annotation segmentation masks from Atari Gopher. We only specify two objects for Gopher. The tunnel in the ground, which is the navigable space for the gopher, is challenging to encode as an object given our model structure.

## J  SAMPLE ANNOTATIONS FOR ATARI, HOLLOW KNIGHT, AND METAWORLD

Figure 13, 12, and 14 present sample frames and annotations used by our method in Atari, Hollow Knight, and Metaworld, respectively. For each task in Atari and Metaworld, we annotate 6 frames, and for each boss in Hollow Knight, we annotate 12 frames.

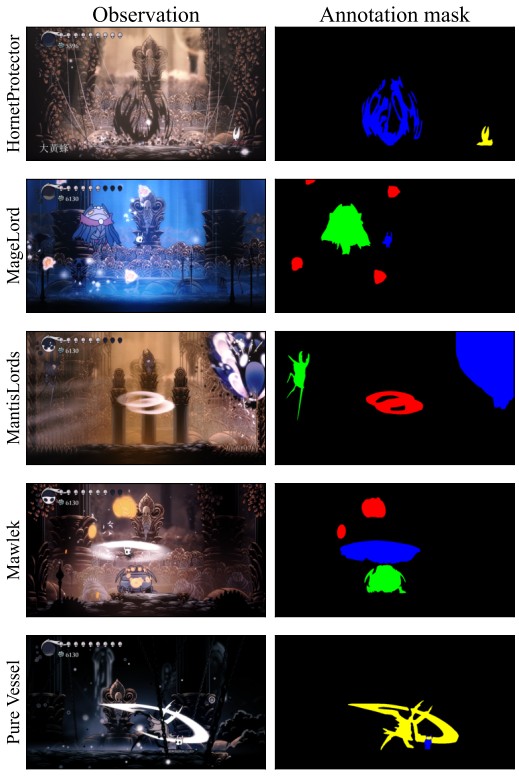

Figure 12: Sample frames and annotation masks for Hollow Knight bosses.

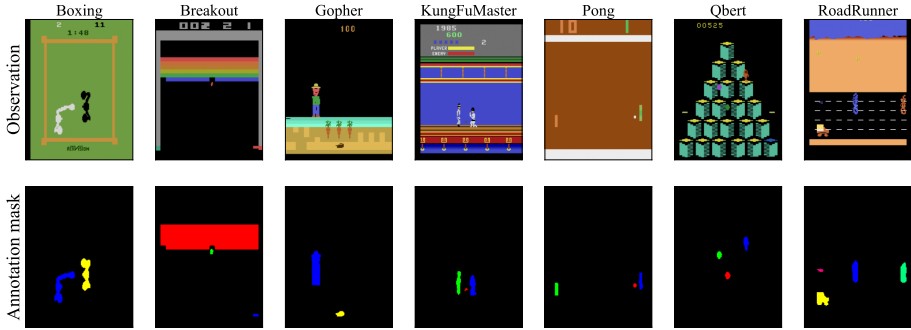

Figure 13: Sample frames and annotation masks for Atari games.

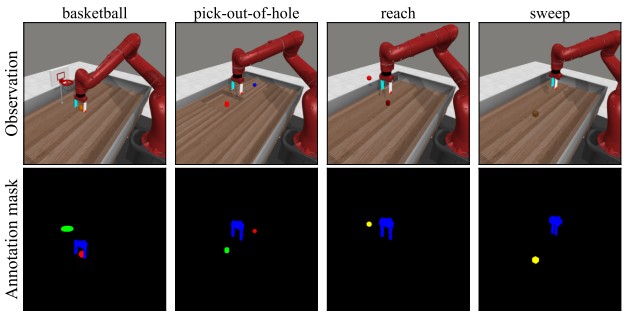

Figure 14: Sample frames and annotation masks for Meta-world tasks.

# K    ABLATION ON MISSING OBJECTS

We conducted an ablation study on removing annotated objects, and all experiments only use object features with no visual input for the world model. We found that:

1. **Missing the object that is directly influenced or controlled by the input actions** causes a significant performance drop. Increasing the number of tracked objects generally improves performance.

2. Some objects (e.g., the brick wall in Breakout) **may negatively affect performance**. We believe this arises from current video segmentation algorithms' limited sensitivity to subtle object-level changes (the entire brick wall is treated as a single object). This limitation is discussed in Appendix I.

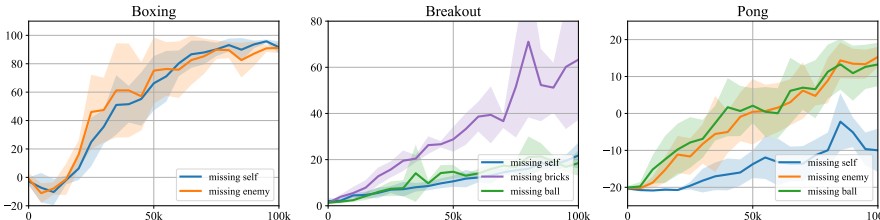

Figure 15: Ablation study on missing objects. Sample game observations can be found in Figure 13.

## L   Unconditional SAM on Hollow Knight

At the beginning of our study, we also explored the possibility of performing segmentation using SAM (Kirillov et al., 2023; Ravi et al., 2024) without any labeled frames. The results, as illustrated in Figure 16, show that SAM can roughly distinguish different objects in the Hollow Knight environment; however, the alignment of semantic meaning and the delineation of object scales were not satisfactory in the absence of prompts. As a result, the quality of the object feature wouldn't be sufficient to support the control objectives of our framework. These observations motivated us to incorporate targeted supervision into our method design.

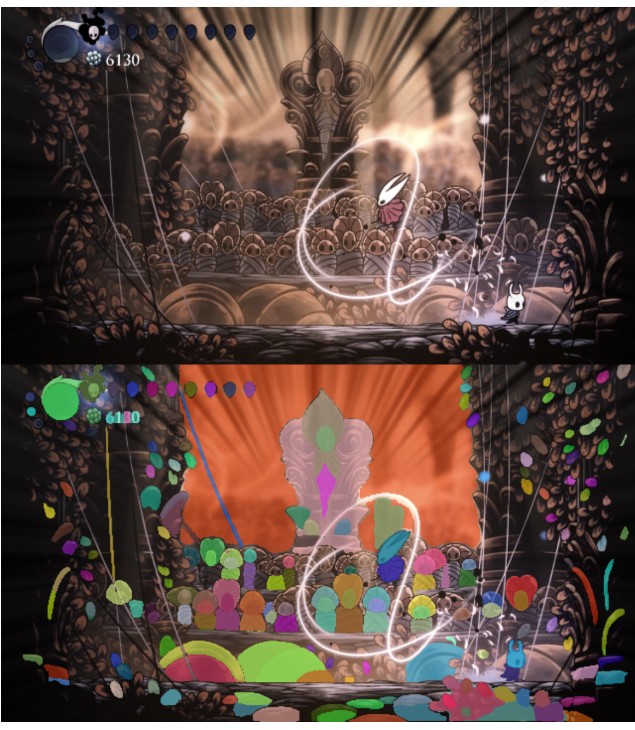

Figure 16: Sample unconditional segmentation results with SAM on Hollow Knight.

## M   The Use of Large Language Models (LLMs)

We use LLMs for polishing and grammar checking. No content is directly generated with LLMs.

