# OpenReview forum: "Object-Centric World Models from Few-Shot Annotations for Sample-Efficient Reinforcement Learning"
_ICLR.cc/2026/Conference — ICLR 2026 Poster_

### Official Review · Reviewer_zM4S · 2025-10-23

**Soundness:** 3
**Presentation:** 2
**Contribution:** 2
**Rating:** 4
**Confidence:** 4

**Summary:**

This paper studies object based representations for learning world models in a sample efficient manner. They propose to use SAM2 and Cutie as the frozen models which provide object vectors. These vectors along with the full frame are passed into categorical vae followed by a spatial transformer whose outputs are used to predict the next latents, rewards and termination. They show the effectiveness of their approach on atari100k, hollow knight and some continuous control tasks.

**Strengths:**

I appreciate the idea of adding object-based inductive biases into world modelling, this allows to maintain object consistency and tracking for all objects especially smaller ones.

I like that the authors opted for wide evaluation suite going beyond atari to hollow knight and continuous control.

**Weaknesses:**

I think the paper would benifit from comparing against unsupervised object-centric representation learning baseline such as slot attention (https://arxiv.org/abs/2006.15055) and slotformer (https://arxiv.org/abs/2210.05861). The main claim of the paper is that object based vectors help world modelling but it is not clear whether there is something special in vectors provides by SAM2/Cutie or even unsupervised methods can also help. Also world modelling has been a focus in the unsupervised object-centric representation learning community (eg. slotformer, SSWM( https://arxiv.org/abs/2402.03326)) hence it might be useful to copmare against some of these baselines.

It would be great if the authors could clarify and make the setup more clear. For example,
- The main model figure is in the appendix, I think it should be in the main paper
- It is not clear how the FOCUS baseline works - how do you discretize the masks from FOCUS?

The claim for discretization is to limit compounding of errors. I don't fully buy this, shouldn't errors compound in discrete autoregressive prediction too? It would be great to have a baseline which does not perform discretization so only continuous prediction to show the pros and cons


The authors claim that the method is sample efficient which is great, however if given access to more samples, will the method scale as well as non object-based methods?

**Questions:**

What is the policy to collect rollouts for training the world model?

How do you decide the number of object vectors for each task?

---

> ### Author Response · Authors · 2025-11-20
> **Part 1 of the initial response**
>
> Thank you for your valuable feedback and for engaging deeply with our work. We address your concerns and questions below.
>
> ## Respond to Weakness
>
> ### **1. Comparison with unsupervised methods**
> **1.1 Comparison with SOLD:** We appreciate the suggestion to include comparisons with unsupervised object-centric approaches. In the updated manuscript, we now provide both qualitative and quantitative results for SOLD (ICML 2025) [1], available in**Appendix M of the updated manuscript (page 30)**. SOLD is a strong unsupervised object‑centric model-based RL control method, unlike many unsupervised OC baselines that are evaluated only on video prediction.
>
> We trained SOLD on the Hollow Knight dataset and several Atari games. Slot-attention–based models struggled with consistent object binding in these highly dynamic environments, even when given substantially more data and compute than OC‑STORM. In terms of episode returns, unsupervised OC features yielded little to no improvement over DreamerV3. We believe this is mainly due to the substantial visual complexity and dynamism of both Atari and Hollow Knight. To our knowledge, existing unsupervised OC approaches have not demonstrated success on Atari, whereas our method **is the first to integrate object‑centric modeling into world models on both the Atari 100k benchmark and the visually challenging Hollow Knight environment**.
>
>
> **1.2 Testing SAM Without Prompts:** We also report results from applying SAM without prompts on Hollow Knight in **Appendix N of the updated manuscript (page 32)**. Without task‑specific priors, object definitions become ambiguous, and even strong open‑set segmentation models fail to produce representations useful for control. This further supports our decision not to adopt unsupervised OC approaches in our work.
>
> ### **2. Main model figure position**
> We agree that presenting the overall setup clearly is important. In the current version, Figure 1 highlights our primary contribution—that we perform object-centric modeling with only a few annotations, rather than fully unsupervised learning.
>
> We placed the full model diagram in the appendix to maintain a clean narrative flow in the main text. This allows readers to understand the conceptual pipeline before diving into technical details, which remain fully documented in the appendix.
>
>
> ### **3. Discretization of latents and the use of FOCUS**
>
> **How discretization is applied:** The discretization operation in our framework is applied in the latent space, after all visual inputs, masks, and object representations are first projected into a shared latent representation. Thus, the masks generated by FOCUS are not directly discretized in pixel space, but rather through quantization in the latent embedding space, consistent with techniques such as Categorical VAE and VQ-VAE.
>
> **On compounding errors:** We agree that discretization does not eliminate error accumulation. However, it substantially reduces the amplification of small prediction deviations that commonly arise in continuous autoregressive models. The table below illustrates this effect: minor continuous errors (e.g., 0.1–0.2) accumulate and drift over time, whereas discretized latent predictions remain within fixed representational boundaries. As noted, large prediction errors still degrade performance in both settings.
>
> | Timestep | 1 (context)     | 2       | 3       | 4       |
> |-------------|---------|---------|---------|---------|
> | Continuous       | | | |
> | Real:    | 0.9   | 0.8     | 1.1     | 0.7     |
> | Model:   | 0.9   | 0.7     | 0.9     | 0.4     |
> | Error:   | 0.0   | 0.1     | 0.2     | 0.3     |
> | Discrete       | | | |
> | Real:    | 3     | 2       | 5       | 1       |
> | Model:   | 3        | 2(2.3)  | 5(4.8)  | 1(1.3)  |
> | Error:   | 0     | 0       | 0       | 0       |
>
>
> To further illustrate this behavior, we provide qualitative and quantitative comparisons between continuous and discretized latent predictions on Atari Boxing in **Appendix P of the update manuscript (page 34)**. The discretized version exhibits notably lower long‑horizon prediction drift, supporting its practical benefit within our world‑modeling framework.
>
> The use of latent discretization follows a long line of established world modeling research [2-5], as well as recent advances in image/video generation tasks [6-7]. In these works, discretized latent spaces were a key factor enabling stable and accurate long-horizon modeling. We follow the same principle here to ensure comparability and methodological consistency.
>
> **Continues on the next comments.**

---

> ### Author Response · Authors · 2025-11-20
> **Part 2 of the initial response**
>
> ### **4. Scaling property on more samples**
>
> Our method is built upon model-based RL frameworks such as DreamerV3 and STORM, which are known to scale well when provided with abundant training data. Since our approach also takes raw visual observation as inputs, it can fully leverage the same learning pipelines as non-object-based baselines. Therefore, in settings with more available samples, our method is **expected to scale at least as well as these baselines**, while retaining its key advantage of higher sample efficiency.
> Moreover, running full 200M-step training on Atari or Minecraft requires at least 7 GPU-days per seed, making such large-scale experiments difficult to conduct during the rebuttal period.
>
> The improved sample efficiency is particularly beneficial in the early stages of training, allowing the agent to encounter higher-quality trajectories much sooner. For example, in the Hollow Knight Hornet task, OC-STORM achieves its first boss defeat **in roughly half the wall-clock time** required by STORM. This difference is meaningful in domains where simulation is slow and in real-world scenarios where data collection is costly (as in Hollow Knight, which runs at original game speed without environment parallelization).
>
> ## Respond to Questions
>
> ### **1. The policy to collect rollouts**
> The world model is trained **fully online**. We improve the world model along with the policy, which is aligned with the settings in previous MBRL work (DreamerV3, STORM, etc.). We alternate one gradient step for the world model with one gradient step for the policy, continually collecting data using the improved policy.
>
> ### **2. Decide the number of objects**
> Since we can combine both low-resolution visual input and object vectors, we can always choose to "under-represent" and only keep the essential objects. We observe consistent performance improvements even when omitting several reward‑related objects, as demonstrated in environments such as Atari Qbert, Krull and Hollow Knight HK Prime.
>
> Based on our experience and the new ablation study added in **Appendix L of the updated manuscript (page 29)**, we provide the following guidelines that consistently work across all tested environments:
>
> 1. Label objects that are directly influenced or controlled by the input actions (these are reliably and unambiguously identifiable in all environments we tested).
>
> 2. Include only the minimal set of objects required to compute the reward.
>
> 3. Exclude objects that cannot be reliably detected by the underlying segmentation method (Cutie or SAM). See the Limitations section (page 9) for details.
>
>
> ---
>
> Thank you again for the detailed and constructive feedback. If our clarifications and additional analyses have addressed your concerns, we would greatly appreciate your consideration in updating your evaluation. We are happy to provide any further experiments or details as needed.
>
>
> ---
> **References**
>
> [1] Malte Mosbach, et al. SOLD: Slot Object-centric Latent Dynamics Models for Relational Manipulation Learning from Pixels. ICML 2025.
>
> [2] Lukasz Kaiser, et al. Model Based Reinforcement Learning for Atari. In 8th International Conference on Learning Representations, ICLR 2020.
>
> [3] Danijar Hafner, et al. Dream to Control: Learning Behaviors by Latent Imagination. ICLR 2020.
>
> [4] Danijar Hafner, et al. Mastering diverse control tasks through world models. Nature 2025.
>
> [5] Vincent Micheli, et al. Transformers are Sample-Efficient World Models. ICLR 2023.
>
> [6] Robin Rombach, et al. High-resolution image synthesis with latent diffusion models. CVPR 2022.
>
> [7] Team Wan, et al. Wan: Open and advanced large-scale video generative models. arXiv 2025.

---

> > ### Comment · Reviewer_zM4S · 2025-11-26
> >
> > I would like to thank the authors for their response and for answering my queries.
> >
> > I appreciate the authors comparing their approach to an unsupervised object-centric method. The authors mentioned that unsupervised methods suffer as they do not work in visually complex environments. While this makes sense for SAVi which is the base for SOLD, recent object-centric methods such as DINOSAUR do work in complex visual domains and it would be great to see a comparison with those.
> >
> > Secondly, I understand that it is hard to conduct experiments regarding scaling the number of samples but I would encourage the authors to include those results later in the paper.
> >
> > However, apart from that I think the authors have addressed my concerns in a satisfactory manner and as a result I would like to raise my score.

---

> > > ### Author Response · Authors · 2025-11-26
> > >
> > > Thank you for your positive feedback and for raising your score. We truly appreciate your recognition of our work.
> > >
> > > We agree that unsupervised OC methods offer valuable flexibility in adapting to novel and irregular objects. In fact, our initial goal was to address the problem of key objects disappearing using unsupervised approaches. However, we found that pretrained supervised approaches are more practical at the current stage, particularly for control tasks. Unsupervised OC methods, while promising, often require substantial data and computing resources that exceed the sample budgets typical of RL benchmarks. We believe that even within the unsupervised OC paradigm, building up generic pretrained models may be necessary to achieve more stable and reliable performance in diverse settings.
> > >
> > > We will include comparisons with DINOSAUR and other recent unsupervised object-centric methods in the revised manuscript. Additionally, we plan to conduct scaling experiments with larger sample budgets in future work, which we agree is an important direction to explore.
> > >
> > > Thank you again for your constructive engagement and support.

---

### Official Review · Reviewer_waoT · 2025-10-31

**Soundness:** 3
**Presentation:** 4
**Contribution:** 2
**Rating:** 6
**Confidence:** 3

**Summary:**

Presents OC-STORM, an object centric model based RL methods:
- The user gives the model a few annotated object masks which are used to get SAM2 / cutie object representations.
- The object vectors and a downsampled image vector are embedded to a categorical latent (via categorical VAE)
- A world model is trained via reconstruction
- A policy is trained inside of the world model for control.

Performance results:
- The object-centric variants of DreamerV3 and STORM tend to perform better than the non-object centric part.
- Further, for games where SAM / Cutie is better at detecting objects in, performance is consistently better, which demonstrates the merits of the method when there is a good object extraction
- Further evaluated in a hollow knight 100k setting and find that it out-performs the on-OC variant
- Can also work in continuous tasks (Metaworld)
- Analysis shows that (i) object information is well-learned as scene can be reconstructed from object vectors (players in boxing), and (ii) there is some robustness to randomly zero-ing out detected objects

**Strengths:**

The idea is novel in using few-shot labelled object masks to get (pre-trained) object representations, which is then used to train a world model for policy control.

The paper presents clearly and is of high quality. It presents positive results against reasonable baselines across a number of environments (atari100k, hollow knight, metaworld). It reads as a comprehensive work that can be informative for future works in object centric RL to build on.

The ablations are comprehensive. The authors do good science to isolate the effect of object encoding (e.g. object detectable vs. badly detected), if the object representation can be reconstructed, and robustness to segmentation model failure.

Finally, I appreciate that authors are also frank in discussing some short-comings of current object extraction methods (such as inability to handle duplicate objects and geometric structures such as walls and floors) for future research to build on.

**Weaknesses:**

The major weakness of this method is the effort vs. gains trade-off for using this object-centric representation. To use OC-STORM, the user must first generate 6-12 frames of object mask labels for each environment they may wish to run. The gain from doing this, based on the paper, seems to be _mainly_ about better _sample efficiency_. One could argue that instead of going through the effort of labelling, the user can also (i) run the alternative methods longer to get similar performance, or (ii) use a more compute-intensive method to get the same sample efficiency (e.g. Delta-IRIS or DIAMOND in Appendix C seem to show this -- correct me if I mis-understood).

On the other hand, some of the original work on object-oriented RL were motivated by the hope that modelling object & interactions can allow generalization in fundamentally different ways [1]. More recent works have similarly shown zero-shot generalization [2,3], new ways of doing model learning & efficient exploration [3], and the ability for compositional generalization [4,5].

If sample efficiency is indeed the main goal, and being mindful that future work can push this method to be even more performant than the more compute intensive methods, it still feels worthwhile to discuss the particular use case for the current approach as presented currently (for instance, in a low sample, low compute setting?), and how it may differ (or is similar) to other object-oriented approaches.

-----

[1] Diuk, Carlos, Andre Cohen, and Michael L. Littman. "An object-oriented representation for efficient reinforcement learning." Proceedings of the 25th international conference on Machine learning. 2008.

[2] Sancaktar, Cansu, Sebastian Blaes, and Georg Martius. "Curious exploration via structured world models yields zero-shot object manipulation." Advances in Neural Information Processing Systems 35 (2022): 24170-24183.

[3] GX-Chen, Anthony, Kenneth Marino, and Rob Fergus. "Efficient Exploration and Discriminative World Model Learning with an Object-Centric Abstraction." arXiv preprint arXiv:2408.11816 (2024).

[4] Zhou, Allan, et al. "Policy architectures for compositional generalization in control." arXiv preprint arXiv:2203.05960 (2022).

[5] Haramati, Dan, Tal Daniel, and Aviv Tamar. "Entity-centric reinforcement learning for object manipulation from pixels." arXiv preprint arXiv:2404.01220 (2024).

**Questions:**

- How do you decide on how many key objects to label in the human annotated frames? Is there any guidance here for what to / not to label?
- Can the segmentation model work without any labelled frames (I was under the impression that SAM can)? If so, how accurate are they at segmenting objects?
- How sensitive is the policy to different human labels? If different users labelled the same game, or labelled a different number of objects, how would this change the segmentation and would the policy still be robust to this?

---

> ### Author Response · Authors · 2025-11-20
> **Part 1 of the initial response**
>
> Thank you for the detailed review and thoughtful questions. We address each point below.
>
> ## Respond to Weaknesses:
>
> ### **1. Comparison with computationally intensive methods like DIAMOND**
> We agree that additional training or heavier computation can improve the performance of pixel‑only approaches. However, this trend highlights a central challenge for scalable world models. For example, DIAMOND outperforms OC‑STORM on Atari, but requires over 10× more computation even at 64×64 resolution. Scaling such methods to higher‑resolution settings (e.g., 256×256) quickly becomes impractical for domains like robotics or drone control, where rapid iteration is essential.
>
> Recent large‑scale work underscores this point. DreamerV4 [1] trained on over 2,500 hours of 640×360 Minecraft data using 1,024 TPU‑v5p chips. Despite this substantial investment, the performance gains relative to the computational cost remain limited. This raises a broader question: Can further scaling of raw pixel models alone produce major breakthroughs, or do we need more structured and efficient representations?
>
> Our work takes a step in the latter direction. Extracting around ten object features from a 640×360 image can provide an effective compression of roughly 30$\times$–270$\times$ (depending on the encoder), enabling more efficient scaling than raw‑pixel approaches. We view OC‑STORM as an early demonstration that lightweight, structured representations can meaningfully improve the cost–efficiency trade‑off in world models.
>
>
> ### **2. Annotation cost**
> Each annotation requires **less than one minute**. Following the Cutie pipeline, we use a semi‑automatic segmentation method based on RITM [1], where masks are generated efficiently by iteratively adding positive and negative anchors. This process yields fast and accurate annotations.
>
> In practice, producing 6–12 annotated frames per environment takes **approximately 10 minutes**. We believe this one‑time cost is modest relative to the overall development cycle of world‑model‑based agents, and the resulting gains in sample efficiency and stability substantially reduce total experimentation time.
>
> ### **3. On not pursuing generalization from object representation in this work**
> Most current object-centric methods are either unsupervised or rely on environment-provided ground truth information, which constrain them to relatively simple or synthetic settings. As discussed in **Appendix M of the updated manuscript (page 30)**, such approaches often fail in complex, visually rich environments, making it unclear how well their reported results on generalization, exploration, or compositionality transfer beyond controlled testbeds.
>
> Our contribution is to integrate few‑shot, pretrained video object understanding models into world models in two visually rich settings: Atari 100k and Hollow Knight. To our knowledge, this is the first demonstration of stable object‑centric world‑model control in environments of this complexity (see supplementary videos). Broader generalization, compositional learning, and exploration-oriented studies are promising future directions that we hope this work will inspire.
>
> **Respond to Questions continued on the next comment.**

---

> ### Author Response · Authors · 2025-11-20
> **Part 2 of the initial response**
>
> ## Respond to Questions:
>
> ### **1. How many objects to label, and what to include?**
> Because OC‑STORM combines low‑resolution visual inputs with object vectors, the representation can be deliberately underspecified. We observe consistent performance gains even when omitting several reward‑related objects (e.g., Atari Qbert, Krull, Hollow Knight HK Prime).
>
> Based on our experience and the ablations added in **Appendix L of the updated manuscript (page 29)**, we provide the following guidelines that consistently work across all tested environments:
>
> 1. Label objects that are directly influenced or controlled by the input actions (these are reliably and unambiguously identifiable in all environments we tested).
>
> 2. Include only the minimal set of objects required to compute the reward.
>
> 3. Exclude objects that cannot be reliably detected by the underlying segmentation method (Cutie or SAM). See the Limitations section (page 9) for details.
>
>
> ### **2. Unconditional segmentation**
> We tried using SAM without any prompts during early experiments. The results were not sufficiently accurate for control, so they were omitted from the initial submission. We have now added these results as **Appendix N in the updated manuscript (page 32)**. Unprompted SAM can roughly distinguish different objects, but the semantic alignment and object‑scale consistency are poor. This makes it unsuitable for downstream world‑model learning.
>
> That said, we believe that with the ongoing progress in vision-language models (VLMs), bridging the gap between game/task-level rules and automatic object extraction will soon become feasible.
>
>
> ### **3. Sensitivity to different human labels**
>
> OC‑STORM shows low sensitivity to variations in human‑provided labels. As noted in our response to Question 1 and demonstrated in **Appendix L (page 29)**, labeling only the controlled object already provides strong performance. Sensitivity is further reduced because OC‑STORM jointly uses object‑level features and raw visual observations. Thus, even if two annotators choose different sets of secondary or contextual objects, the visual backbone compensates for these differences, resulting in stable segmentation and robust policy performance.
>
>
> ---
>
> We hope these clarifications address your concerns. If our responses and additional analyses satisfactorily resolve your questions, we kindly ask you to consider reflecting this in your assessment. We would be glad to provide further details if needed.
>
> ---
>
>
> **References**
>
> [1] Danijar Hafner, et al. Training agents inside of scalable world models. arXiv 2025.
>
> [2] Konstantin Sofiiuk, et al. Reviving iterative training with mask guidance for interactive segmentation. 2022 IEEE international conference on image processing (ICIP). IEEE, 2022.

---

### Official Review · Reviewer_FttL · 2025-11-01

**Soundness:** 3
**Presentation:** 2
**Contribution:** 3
**Rating:** 6
**Confidence:** 3

**Summary:**

The paper proposes OC-STORM, an object-centric model-based RL framework that fuses few-shot, pretrained video-segmentation features with pixel inputs to train a spatial-temporal world model, yielding improved sample efficiency on Atari-100k and Hollow Knight.

**Strengths:**

- The method integrates few-shot object features from SAM2/Cutie into a spatial–temporal world model (Transformer/RNN backbones), with clean modality separation (object tokens + visual token) and categorical VAE discretization; the training and architecture are well specified.

- On Atari-100k, object-centric variants outperform baselines (e.g., Cutie-OC-STORM reaches HNS mean 134.8% vs. STORM 114.2%, median 43.8% vs. 42.5%); Hollow Knight learning curves show faster convergence on harder bosses.

- The paper diagnoses why vector features beat mask features, provides module ablations, feature-only reconstructions, and a failure-robustness study by zeroing object features; Meta-World results indicate portability beyond Atari.

**Weaknesses:**

- Comparative scope. Core comparisons are mainly within-framework ablations (STORM/DreamerV3 variants); external SOTA world-model baselines (e.g., diffusion/tokenization variants) and broader agent baselines are deferred or absent in the main text, and Hollow Knight lacks standardized settings—making cross-paper claims harder to calibrate.

- Annotation/K configuration burden. The user-set K (objects) and handful of annotated frames (≈6–12) are reasonable but the human-time budget and sensitivity to K are not quantified in the main text; detection incompleteness is acknowledged but only coarsely analyzed.

- System metrics incomplete. The paper references computational overhead analyses but does not report build/index/latency figures for segmentation + KG-like memory in the main body—useful for scaling to long episodes or high-res inputs.

**Questions:**

- Labeling cost & K sensitivity. How many minutes of annotation per game are required in practice, and how does performance vary with K (under-/over-specifying the number of tracked objects)? Could you add a curve for returns vs. K and vs. number of annotated frames?

- External baselines & reporting. Can you include a matched-config comparison against recent token/diffusion world models and standardized Hollow Knight setups (or release your wrapper to make one), plus runtime/latency tables for the segmentation pipeline?

---

> ### Author Response · Authors · 2025-11-20
>
> Thank you for the detailed feedback and for the opportunity to clarify our contributions. We address each concern below.
>
> ## Respond to Weaknesses and Questions
>
> ### **1. Annotation cost**
> Each annotation requires **less than one minute**. Following the Cutie pipeline, we use a semi‑automatic segmentation method based on RITM [1], where masks are generated efficiently by iteratively adding positive and negative anchors. This process yields fast and accurate annotations.
>
> In practice, producing 6–12 annotated frames per environment takes **approximately 10 minutes**. We believe this one‑time cost is modest relative to the overall development cycle of world‑model‑based agents, and the resulting gains in sample efficiency and stability substantially reduce total experimentation time.
>
>
> ### **2. Annotation sensitivity**
>
> **2.1 Under-/Over-specifying:** We include new experiments analyzing the "return vs. K" curve. The results are presented in **Appendix L of the updated manuscript (page 29)**. Our method is generally robust to both under‑ and over‑specification, with two key findings:
>
> 1. Missing the object that is directly influenced or controlled by the input actions causes a significant performance drop. Increasing the number of tracked objects generally improves performance.
> 2. Including some objects (e.g., the brick wall in Breakout) may negatively affect performance. We believe this arises from Cutie’s limited sensitivity to subtle object-level changes (the entire brick wall is treated as a single object). This limitation is discussed in Appendix I (page 26).
>
> These experiments use only object features. In practice, when visual input is retained, one can safely choose a smaller K and keep only the well-defined and most influential objects.
>
>
> **2.2 Number of annotated frames:** We already analyzed "returns vs. the number of annotated frames" in **Appendix F.2 (pages 22–23)**. Using 6 annotated frames stabilizes training, but even a single annotated frame is often sufficient. When visual observations are included (the full method), a single annotated frame would provide noticeable improvement.
>
> Multiple annotations are useful because objects may vary in color, pose, or configuration across the episode, and some objects may not appear simultaneously. Annotating several frames ensures full coverage of these variations.
>
>
> ### **3. Comparison with token/diffusion-based world models**
> Our goal is to evaluate the benefits of object‑centric modeling within world models, rather than to exhaustively benchmark against all recent architectures. Including comprehensive cross‑model comparisons in the main text would dilute the central contribution: object‑centric representations, when paired with pretrained foundation models, provide consistent improvements across world‑model designs.
>
> Token‑ and diffusion‑based world models also require substantially higher computational resources. Running matched‑configuration experiments for such systems is not feasible within the rebuttal period ($\approx$3 GPU days per successful run). Importantly, **nothing in these architectures conflicts with object‑centric modeling**. In principle, our representation can be integrated directly into token‑ or diffusion‑based approaches (e.g., IRIS, DIAMOND) via standard attention mechanisms.
>
> ### **4. Standardize Hollow Knight setups**
>
> Our supplementary code already includes **all necessary wrappers** for interacting with the game, and we will release the full repository. On the game side, we provide a custom MOD that logs hit and damage events for our RL interface. A unified configuration is ensured through a provided save file, and full environment details appear in Appendix E.3 (page 19). These resources will be made publicly available.
>
> Due to copyright restrictions, we cannot release the entire game as a packaged environment. However, the provided wrappers, configuration files, and logging MOD enable full reproducibility with minimal setup.
>
> ### **5. Runtime latency**
>
> We have added runtime and latency measurements in **Appendix O of the updated manuscript (page 33)**. The segmentation pipeline introduces only modest computational overhead and remains compatible with real‑time control.
>
> ---
>
> We hope these clarifications address your concerns. If our responses and additional analyses satisfactorily resolve your questions, we kindly ask you to consider reflecting this in your assessment. We would be glad to provide further details if needed.
>
> ---
> **References**
>
> [1] Konstantin Sofiiuk, et al. Reviving iterative training with mask guidance for interactive segmentation. 2022 IEEE international conference on image processing (ICIP). IEEE, 2022.

---

### Author Response · Authors · 2025-12-01

Dear AC,

Thank you for your extra work in this special situation. We understand the workload is significant. Below is a brief summary of our discussion process:

1. During the discussion period, one reviewer (zM4S) indicated that most of their concerns were addressed and adjusted their score from 4 to 6, stating:
   > ..., apart from that I think the authors have addressed my concerns in a satisfactory manner and as a result I would like to raise my score.

2. The other two reviewers (FttL and waoT)  did not respond during the discussion period. We have provided detailed responses to all weaknesses and questions they raised.

3. During the rebuttal, we added four new groups of experiments and five new sections to the appendix to provide additional empirical support for our responses.

We are happy to provide any clarification or additional information if needed.

Best regards,

Authors of Submission 6126

---

### Meta-Review · Area_Chair_bszZ · 2026-01-09

**Summary:**

This paper proposes OC-STORM, an object-centric world model for reinforcement learning. The method leverages object-centric representations obtained from a pre-trained segmentation model. With a minimal number of annotated frames, OC-STORM learns decision-relevant object dynamics and inter-object interactions. Empirical results demonstrate its effectiveness on the standard Atari100k benchmark as well as on a newly introduced, more complex environment, Hollow Knight.

Overall, the reviewers appreciate the sound methodological design, strong empirical performance, and extensive evaluation across multiple benchmarks, including thorough ablation studies. The primary concern raised across reviews relates to the cost of annotations, particularly sensitivity to different human labels and the number of tracked objects. This concern is closely tied to the necessity of comparison with approaches based on unsupervised object representations.

**Reviewer Concerns:**

The major technical concerns regarding the design choice of few-shot annotations have been adequately addressed during the rebuttal. However, the presentation in the main text could be improved to more explicitly clarify these concerns and design decisions.

**Reviewer Scores:**

Reviewer FttL did not respond after the rebuttal. The listed weaknesses and questions were generally well addressed, and the clarification of annotation costs was satisfactory. In the AC’s opinion, comparisons with state-of-the-art world models are largely orthogonal to the core contributions of this paper. The reviewer is likely to maintain their positive score.

Reviewer waoT also did not respond. Their concerns largely overlap with those of Reviewer FttL. The additional comparisons with unsupervised segmentation and object-centric representations help justify the design choice of using few-shot annotations. This reviewer is also likely to maintain a positive score.

Reviewer zM4S raised similar concerns and adjusted the rating to 6 prior to the major leakage issue.

---

### Decision · Program_Chairs · 2026-01-26

Accept (Poster)